# Efficient Parametric SVD of Koopman Operator for Stochastic Dynamical Systems

**Minchan Jeong**[1]*, **J. Jon Ryu**[2]*, **Se-Young Yun**[1], **Gregory W. Wornell**[2]
[1]Kim Jaechul Graduate School of AI, KAIST, Daejeon 34141, South Korea
[2]Department of EECS, MIT, Cambridge, MA 02139, United States
{mcjeong,yunseyoung}@kaist.ac.kr, {jongha,gww}@mit.edu

## Abstract

The Koopman operator provides a principled framework for analyzing nonlinear dynamical systems through linear operator theory. Recent advances in dynamic mode decomposition (DMD) have shown that trajectory data can be used to identify dominant modes of a system in a data-driven manner. Building on this idea, deep learning methods such as VAMPnet and DPNet have been proposed to learn the leading singular subspaces of the Koopman operator. However, these methods require backpropagation through potentially numerically unstable operations on empirical second moment matrices, such as singular value decomposition and matrix inversion, during objective computation, which can introduce biased gradient estimates and hinder scalability to large systems. In this work, we propose a scalable and conceptually simple method for learning the top-$k$ singular functions of the Koopman operator for stochastic dynamical systems based on the idea of low-rank approximation. Our approach eliminates the need for unstable linear-algebraic operations and integrates easily into modern deep learning pipelines. Empirical results demonstrate that the learned singular subspaces are both reliable and effective for downstream tasks such as eigen-analysis and multi-step prediction.

## 1 Introduction

The Koopman operator theory offers a powerful framework for analyzing nonlinear dynamical systems by lifting them into an infinite-dimensional function space, where spectral techniques from linear operator theory can be applied. Recent advances in dynamic mode decomposition (DMD) have shown that trajectory data can be effectively used to identify dominant eigen-modes in a data-driven manner [40, 46, 47, 18, 4]. Inspired by the success of deep learning, recent methods such as VAMPnet [48, 24] and DPNet [16] employ neural networks to approximate the leading singular subspaces of the Koopman operator. While shown effective for some benchmark tasks, these approaches often rely on numerically unstable operations such as singular value decomposition (SVD) or matrix inversion during objective computation. These operations present practical challenges, particularly in computing unbiased gradients and thus scaling to high-dimensional systems.

In this work, we propose a conceptually simple and scalable method for learning the top-$k$ singular functions of the Koopman operator for stochastic dynamical systems. Our approach builds on the idea of *low-rank approximation*, which has recently received increasing attention in the literature due to its favorable optimization structure that aligns well with modern optimization practices; see, e.g., [21, 45] in the numerical optimization literature, and [42, 39, 8, 32, 49, 17] in the machine learning literature. Our method avoids unstable linear-algebraic computations and is easy to integrate into modern deep learning pipelines. We demonstrate that it reliably recovers dominant Koopman eigen-subspaces and supports downstream tasks such as prediction and eigen-analysis.

---

*The two authors contributed equally to this work.

39th Conference on Neural Information Processing Systems (NeurIPS 2025).

## 2 Problem Setting and Preliminaries

In this section, we introduce the problem setting and establish the foundation for the subsequent discussion. We begin by formulating the problem in the context of discrete-time dynamical systems, which will serve as our primary focus. We then briefly address the continuous-time case. Next, we review two existing approaches, VAMPnet and DPNet, and highlight their limitations, thereby motivating the need for our proposed method.

**Notation.** We denote linear operators by stylized script letters, e.g., $\mathcal{U}$, $\mathcal{K}$, $\mathcal{L}$, and $\mathcal{I}$, using the Zapf Chancery font to distinguish them from matrices and scalar functions. Bold lowercase letters such as $\mathbf{x}$ and $\mathbf{y}$ are reserved for vectors, while bold uppercase letters like $\mathbf{X}$ denote vector-valued random variables. Sans-serif uppercase letters, e.g., M and S, are used for matrices. In particular, I denotes the identity matrix.

### 2.1 Discrete-Time Dynamical Systems

Consider a stochastic discrete-time dynamical system $\mathbf{x}_{t+1} = \xi(F(\mathbf{x}_t), \epsilon_t)$. Here, $F \colon \mathcal{X} \to \mathcal{X}$ is a possibly nonlinear mapping for a domain $\mathcal{X} \subseteq \mathbb{R}^d$, $\epsilon_t \sim \mathcal{D}$ is an independent random noise variable, and $\mathbf{x}_t \mapsto \xi(F(\mathbf{x}_t), \epsilon_t) \in \mathcal{X}$ captures independent noise in the process, such as the additive white Gaussian noise.[2] Assuming that $F$ and the noise distribution $\mathcal{D}$ are time-invariant, the process becomes a time-homogeneous Markov process with transition density $p(\mathbf{x}' \,|\, \mathbf{x})$, i.e., $\Pr(\mathbf{X}_{t+1} \in A \,|\, \mathbf{X}_t = \mathbf{x}) = \int_A p(\mathbf{x}' \,|\, \mathbf{x}) \, d\mathbf{x}'$ for all measurable sets $A \subseteq \mathcal{X}$ and all $t \geq 0$.

In the stochastic setup, the dynamics is fully captured by the transition density $p(\mathbf{x}' \,|\, \mathbf{x})$, which is induced by $(\xi \circ F)(\cdot)$, and thus the problem becomes analyzing $p(\mathbf{x}' \,|\, \mathbf{x})$ of a Markov chain. Here, the Koopman operator becomes the conditional expectation operator, i.e., for an observable $g \colon \mathcal{X} \to \mathbb{R}$,

$$(\mathcal{K}g)(\mathbf{x}) \triangleq \mathbb{E}_{p(\mathbf{x}' \,|\, \mathbf{x})}[g(\mathbf{x}')].$$

By the Markov property, repeatedly applying the Koopman operator will correspond to the *multi-step prediction* in terms of the posterior mean of $g(\mathbf{X}_t)$ given $\mathbf{X}_0 = \mathbf{x}_0$, i.e., $(\mathcal{K}^t g)(\mathbf{x}_0) = \mathbb{E}_{p(\mathbf{x}_t \,|\, \mathbf{x}_0)}[g(\mathbf{x}_t)]$.

Throughout the paper, we will assume that $\mathcal{K}$ is *compact* following [16], which is a mild assumption that holds for a large class of Markov processes; see, e.g., [14]. We remark that we cannot directly apply the existing spectral techniques to a deterministic dynamical system, including the technique developed in this paper, since the corresponding Koopman operator is not compact; see [48, Appendix A.5]. The consideration of stochasticity breaks the degeneracy and allows the linear algebraic techniques to be applicable. Moreover, assuming stochasticity is not necessarily a restrictive assumption, as physical processes in the real world may be inherently noisy.

**Problem Setting.** Our goal is to analyze the Markov dynamics of the length-$T$ trajectory $(\mathbf{x}_0, \mathbf{x}_1, \ldots, \mathbf{x}_T) \sim p(\mathbf{x}_0)p(\mathbf{x}_1 \,|\, \mathbf{x}_0) \ldots p(\mathbf{x}_T \,|\, \mathbf{x}_{T-1})$, where $p(\mathbf{x}_0)$ is a distribution for the initial state. In practice, we assume that we have access to $N$ independent, random trajectories, and collect the consecutive transition pairs $\{(\mathbf{x}_0, \mathbf{x}_1), (\mathbf{x}_1, \mathbf{x}_2), \ldots, (\mathbf{x}_{T-1}, \mathbf{x}_T)\}$ and define the joint distribution over the pair as $p(\mathbf{x}, \mathbf{x}') \triangleq \frac{1}{T} \sum_{t=0}^{T-1} p_t(\mathbf{x})p(\mathbf{x}' \,|\, \mathbf{x})$, where $p_0(\mathbf{x}) = p(\mathbf{x}_0)$ and $p_{t+1}(\mathbf{x}') \triangleq \int p(\mathbf{x}' \,|\, \mathbf{x})p_t(\mathbf{x})d\mathbf{x}$. We let $\rho_0(\mathbf{x})$ and $\rho_1(\mathbf{x}')$ denote the marginal distribution over $\mathcal{X}$ of the current and future states. Note that while $\rho_1(\mathbf{x}') = \mathbb{E}_{\rho_0(\mathbf{x})}[p(\mathbf{x}' \,|\, \mathbf{x})]$ holds, the two distributions, $\rho_0(\mathbf{x})$ and $\rho_1(\mathbf{x}')$ can differ significantly, particularly when the trajectory length $T$ is relatively short.

Let $(\mathcal{X}, \mathcal{F})$ be a measurable space and let $\rho_0$ and $\rho_1$ be (finite) measures on $(\mathcal{X}, \mathcal{F})$ representing the distributions of the current and future states, respectively. For any measure $\rho$ on $(\mathcal{X}, \mathcal{F})$ define $L^2_\rho(\mathcal{X}) \triangleq \{f \colon \mathcal{X} \to \mathbb{C} | \int |f(\mathbf{x})|^2 \rho(d\mathbf{x}) < \infty\}$. Equipped with the inner product $\langle f, g \rangle_\rho \triangleq \int f(\mathbf{x})g(\mathbf{x})\rho(d\mathbf{x})$, $L^2_\rho(\mathcal{X})$ is a Hilbert space. The Koopman operator $\mathcal{K}$ is then a mapping from $L^2_{\rho_1}(\mathcal{X})$ to $L^2_{\rho_0}(\mathcal{X})$, which is an integral kernel operator with the transition kernel $k(\mathbf{x}, \mathbf{x}') \triangleq \frac{p(\mathbf{x}' \,|\, \mathbf{x})}{\rho_1(\mathbf{x}')}$. The adjoint operator $\mathcal{K}^*$ then acts as the *backward predictor*, i.e., $(\mathcal{K}^* f)(\mathbf{x}') = \mathbb{E}_{q(\mathbf{x} \,|\, \mathbf{x}')}[f(\mathbf{x})]$, where $q(\mathbf{x} \,|\, \mathbf{x}') \triangleq \frac{\rho_0(\mathbf{x})p(\mathbf{x}' \,|\, \mathbf{x})}{\rho_1(\mathbf{x}')}$ denotes the conditional distribution induced by $\rho_0(\mathbf{x})p(\mathbf{x}' \,|\, \mathbf{x})$. Similar to $\mathcal{K}$, repeated application of $\mathcal{K}^*$ yields multi-step backward prediction: $((\mathcal{K}^*)^t f)(\mathbf{x}_0) = \mathbb{E}_{p(\mathbf{x}_0 \,|\, \mathbf{x}_t)}[f(\mathbf{x}_0)]$.

---

[2]All machinery developed here can also be adapted for a discrete state space, but we focus on continuous $\mathcal{X}$.

**Special Case 1: Stationary Markov Processes.** Given $p(\mathbf{x}' \,|\, \mathbf{x})$, let $\pi(\mathbf{x})$ denote the stationary distribution, i.e., the distribution satisfies $\mathbb{E}_{\pi(\mathbf{x})}[p(\mathbf{x}' \,|\, \mathbf{x})] = \pi(\mathbf{x}')$. Under mild regularity conditions such as ergodicity and irreducibility, such distribution uniquely exists.[3] In the stationary case, the marginal distributions of current and future states are equal, i.e., $\rho_0 = \rho_1 = \pi$, and the Koopman operator $\mathcal{K}$ becomes a map from $L^2_\pi(\mathcal{X})$ to $L^2_\pi(\mathcal{X})$. Assuming ergodicity, we can collect the time-lagged pairs $(\mathbf{x}, \mathbf{x}')$ from a long, single trajectory, as time averages converge to expectations under the stationary distribution.

**Special Case 2: Reversible Markov Processes.** A Markov process is *time-reversible* if and only if it satisfies the *detailed balance condition*: the joint function $P(\mathbf{x}, \mathbf{x}') \triangleq \pi(\mathbf{x})p(\mathbf{x}' \,|\, \mathbf{x})$ is symmetric, i.e., $P(\mathbf{x}, \mathbf{x}') = P(\mathbf{x}', \mathbf{x})$ for any $\mathbf{x}, \mathbf{x}' \in \mathcal{X} \times \mathcal{X}$. In this case, the Koopman operator becomes self-adjoint, and thus the eigenfunctions and eigenvalues are real. This is a much stronger condition than the normality of a Koopman operator and thus the easiest to deal with. For the case of reversible processes, Noé and Nüske [28] proposed a method to approximate eigenfunctions from time-series data, followed by the extended DMD (EDMD) [46].

## 2.2 Continuous-Time Dynamical Systems

Let $(\mathbf{x}_t)_{t \geq 0}$ be a time-homogeneous Markov process (e.g., Langevin dynamics). The *Koopman semigroup* $\{\mathcal{U}^t\}_{t \geq 0}$ acts on observables $f\colon \mathcal{X} \to \mathbb{R}$ as $(\mathcal{U}^t f)(\mathbf{x}_0) \triangleq \mathbb{E}_{p(\mathbf{x}_t \,|\, \mathbf{x}_0)}[f(\mathbf{x}_t)]$. This defines a strongly continuous semigroup satisfying $\mathcal{U}^0 = I$ (identity operator) and $\mathcal{U}^t \mathcal{U}^s = \mathcal{U}^{t+s}$. The *(infinitesimal) generator* $\mathcal{L}$ of the Koopman semigroup is given by:

$$(\mathcal{L}f)(\mathbf{x}) \triangleq \lim_{t \to 0} \frac{(\mathcal{U}^t f)(\mathbf{x}) - f(\mathbf{x})}{t} = \frac{d}{dt}\mathbb{E}_{p(\mathbf{x}_t \,|\, \mathbf{x}_0)}[f(\mathbf{x}_t)],$$

where the limit is taken in the strong operator topology.

In general, generators may not be bounded, and thus not compact. Similar to [48], however, one can view that this discrete-time dynamics is a discretized version of a continuous dynamics with lag time $\tau > 0$, and the Koopman operator of the discretized system is often compact; see [48]. Moreover, as argued in Kostic et al. [16], we can also directly apply the developed technique for special, yet important continuous-time dynamical systems such as an overdamped Langevin dynamics. Unless stated explicitly (like in Section 3.1), we will describe our techniques for discrete-time processes.

## 2.3 Data-Driven Learning of Singular Subspaces: VAMPnet and DPNet

Learning the dominant singular subspaces of the Koopman operator is a key goal for understanding complex dynamical systems, especially in data-driven modeling. First, for non-normal operators, which commonly arise in irreversible or non-equilibrium dynamics, the singular value decomposition (SVD) provides the best possible low-rank approximation measured by Hilbert–Schmidt norm. Second, if a process is reversible and stationary (i.e., $\rho_0(\mathbf{x}) = \rho_1(\mathbf{x}) = \pi(\mathbf{x})$), the Koopman operator is self-adjoint. In this case, its SVD coincides with its eigenvalue decomposition (EVD). Third, even for irreversible processes, the dominant singular functions still capture essential dynamical features. For example, they can show kinetic distances between states, much like eigenfunctions do in reversible systems [31], or also find coherent sets in changing Markov processes; these are generalized forms of long-lasting states [13]. See also [48, Section 2.3].

These theoretical advantages have motivated recent developments in neural network-based approaches, such as VAMPnet [48, 24] and DPNet [16], which aim to learn the top-$k$ singular subspace of the Koopman operator directly from trajectory data. Both methods are grounded in variational characterizations of the dominant singular subspaces, yet they differ significantly in their training objectives and optimization strategies. In the following, we briefly contrast these training approaches and highlight their common limitations. Here we focus on the training approaches, and defer the discussion on their inference procedures to Section 3.2. Also, unlike the score convention commonly used in the literature, we will follow the *loss* convention throughout, whereby the training objective is formulated as a quantity to be minimized.

---

[3]For example, if the stochasticity in the system is induced by an additive noise, i.e., $\mathbf{x}_{t+1} = F(\mathbf{x}_t) + \epsilon_t$ for $\epsilon_t \sim \mathcal{D}$, with the density of $\mathcal{D}$ being positive almost everywhere, then the system is asymptotically stable, i.e., it converges to a unique stationary distribution; see [19, Corollary 10.5.1].

### 2.3.1 Common Setup

Suppose that we wish to capture the top-$k$ singular subspaces using neural networks $\mathbf{x} \mapsto \mathbf{f}(\mathbf{x}) \triangleq [f_1(\mathbf{x}), \ldots, f_k(\mathbf{x})]^\mathsf{T} \in \mathbb{R}^k$ and $\mathbf{x}' \mapsto \mathbf{g}(\mathbf{x}') \triangleq [g_1(\mathbf{x}'), \ldots, g_k(\mathbf{x}')]^\mathsf{T} \in \mathbb{R}^k$. These are sometimes referred to as the *encoder* and the *lagged encoder*, respectively, which are intended to capture the left and right singular subspaces. Since a Koopman operator, as a special example of canonical dependence kernels, always has the top singular functions $\phi_1(\mathbf{x}) \equiv 1$ and $\psi_1(\mathbf{x}') \equiv 1$ with singular value 1, we can simply set $f_1(\mathbf{x}) \leftarrow 1$ and $g_1(\mathbf{x}') \leftarrow 1$ [32]. We refer to this operation as *centering*, as it ensures that the remaining modes are *centered*, i.e., orthogonal to the constant modes. We adopt this parameterization by default, although many existing methods do not follow this practice. Properly handling the constant modes is crucial, as mishandling them can lead to misleading results; see Appendix A.2 for further discussion.

In a variational optimization framework for SVD, the key quantities are *second-moment matrices*. We thus introduce the following shorthand notation. The *second-moment matrix* of vector-valued functions $\mathbf{h}_1(\cdot)$ and $\mathbf{h}_2(\cdot)$ with respect to a distribution $\rho$ is denoted as

$$\mathsf{M}_\rho[\mathbf{h}_1, \mathbf{h}_2] \triangleq \mathbb{E}_{\rho(\mathbf{x})}[\mathbf{h}_1(\mathbf{x})\mathbf{h}_2(\mathbf{x})^\mathsf{T}],$$

and we write $\mathsf{M}_\rho[\mathbf{h}] \triangleq \mathsf{M}_\rho[\mathbf{h}, \mathbf{h}]$ for shorthand. The *joint second-moment matrix* of $\mathbf{h}_1(\cdot)$ and $\mathbf{h}_2(\cdot)$ over the joint distribution $\rho_0(\mathbf{x})p(\mathbf{x}' \mid \mathbf{x})$ is defined and denoted as

$$\mathsf{T}[\mathbf{h}_1, \mathbf{h}_2] \triangleq \mathbb{E}_{\rho_0(\mathbf{x})p(\mathbf{x}' \mid \mathbf{x})}[\mathbf{h}_1(\mathbf{x})\mathbf{h}_2(\mathbf{x}')^\mathsf{T}],$$

which satisfies the identities $\mathsf{T}[\mathbf{h}_1, \mathbf{h}_2] = \mathsf{M}_{\rho_0}[\mathbf{h}_1, \mathcal{K}\mathbf{h}_2] = \mathsf{M}_{\rho_1}[\mathcal{K}^*\mathbf{h}_1, \mathbf{h}_2]$.

In what follows, we will define a *population* objective function using the population second moment matrices $\mathsf{M}_{\rho_0}[\mathbf{f}]$, $\mathsf{M}_{\rho_1}[\mathbf{g}]$, and $\mathsf{T}[\mathbf{f}, \mathbf{g}]$. In practice, these quantities are to be estimated with trajectories, and their empirical estimates are denoted with hats (ˆ), i.e., $\hat{\mathsf{M}}_{\rho_0}[\mathbf{f}]$, $\hat{\mathsf{M}}_{\rho_1}[\mathbf{g}]$, and $\hat{\mathsf{T}}[\mathbf{f}, \mathbf{g}]$.

### 2.3.2 VAMPnet

For an integer $r \geq 1$, Wu and Noé [48] introduce the *VAMP-r objective*[4]

$$\mathcal{L}_{\mathsf{vamp}\text{-}r}(\mathbf{f}, \mathbf{g}) \triangleq -\left\|(\mathsf{M}_{\rho_0}[\mathbf{f}])^{-\frac{1}{2}}\mathsf{T}[\mathbf{f}, \mathbf{g}](\mathsf{M}_{\rho_1}[\mathbf{g}])^{-\frac{1}{2}}\right\|_r^r. \tag{1}$$

Here, $\|\cdot\|_r$ denotes the Schatten $r$-norm. The variational principle behind the VAMP-$r$ objective is explained in Appendix B.1 for completeness. In practice with finite samples, to avoid the numerical instability in computing the inverse matrices, $\hat{\mathcal{L}}_{\mathsf{vamp}\text{-}r}^{(\lambda)}(\mathbf{f}, \mathbf{g}) \triangleq \|(\hat{\mathsf{M}}_{\rho_0}[\mathbf{f}] + \lambda\mathsf{I})^{-\frac{1}{2}}\hat{\mathsf{T}}[\mathbf{f}, \mathbf{g}](\hat{\mathsf{M}}_{\rho_1}[\mathbf{g}] + \lambda\mathsf{I})^{-\frac{1}{2}}\|_r^r$. Tuning $\lambda$ in practice can be done by a cross validation. In the literature, the use of $r = 1$ [24] or $r = 2$ [48] has been advocated.

### 2.3.3 DPNet

Kostic et al. [16] proposed an alternative objective, called the *deep projection* (DP) objective,

$$\mathcal{L}_{\mathsf{dp}}^{(\gamma)}(\mathbf{f}, \mathbf{g}) \triangleq -\left\|(\mathsf{M}_{\rho_0}[\mathbf{f}])^{-\frac{1}{2}}\mathsf{T}[\mathbf{f}, \mathbf{g}](\mathsf{M}_{\rho_1}[\mathbf{g}])^{-\frac{1}{2}}\right\|_{\mathrm{F}}^2 + \gamma(\mathcal{R}(\mathsf{M}_{\rho_0}[\mathbf{f}]) + \mathcal{R}(\mathsf{M}_{\rho_1}[\mathbf{g}])), \tag{2}$$

where they further introduced the metric distortion loss $\mathcal{R} \colon \mathbb{S}_+^L \to \mathbb{R}_+$ defined as $\mathcal{R}(\mathsf{M}) \triangleq \mathrm{tr}(\mathsf{M}^2 - \mathsf{M} - \ln \mathsf{M})$ for $\mathsf{M} \succeq 0$. Note that with $\gamma = 0$, the objective becomes equivalent to the VAMP-2 objective. To detour the potential numerical instability of the first term, which is the VAMP-2 objective, the authors further proposed a relaxed objective called the DP-relaxed objective

$$\mathcal{L}_{\mathsf{dpr}}^{(\gamma)}(\mathbf{f}, \mathbf{g}) \triangleq -\frac{\|\mathsf{T}[\mathbf{f}, \mathbf{g}]\|_{\mathrm{F}}^2}{\|\mathsf{M}_{\rho_0}[\mathbf{f}]\|_{\mathrm{op}}\|\mathsf{M}_{\rho_1}[\mathbf{g}]\|_{\mathrm{op}}} + \gamma(\mathcal{R}(\mathsf{M}_{\rho_0}[\mathbf{f}]) + \mathcal{R}(\mathsf{M}_{\rho_1}[\mathbf{g}])), \tag{3}$$

where $\|\cdot\|_{\mathrm{op}}$ denotes the operator norm. In both cases, Kostic et al. [16] argued that using $\gamma > 0$ is crucial for improving the quality of the learned subspaces. In contrast, the scheme we propose below does not require such regularization, and we empirically found that it offers no benefit.

---

[4]We note that this expression corresponds to the *maximal* VAMP-$r$ score in the original paper [48], but we call this VAMP-$r$ objective, which is a slight abuse, as this is the training objective to train neural network basis.

## 2.4 Practical Limitations of VAMPnet and DPNet

Although these objectives are well-founded in the population limit for characterizing the desired singular subspaces, they do not permit efficient optimization via modern mini-batch-based training.

To compute the VAMP-1 objective [24], one must evaluate the matrix square root inverse of $\hat{\mathsf{M}}_{\rho_0}[\mathbf{f}]$ and $\hat{\mathsf{M}}_{\rho_1}[\mathbf{g}]$, as well as the nuclear norm of the matrix $(\hat{\mathsf{M}}_{\rho_0}[\mathbf{f}])^{-1/2}\hat{\mathsf{T}}[\mathbf{f},\mathbf{g}](\hat{\mathsf{M}}_{\rho_1}[\mathbf{g}])^{-1/2}$. These computations involve numerical linear algebra operations such as eigenvalue decomposition and singular value decomposition of empirical second-moment matrices. Such inverses may be ill-defined or numerically unstable when $\mathrm{span}(\mathbf{f})$ is nearly rank-deficient during optimization, and further, backpropagating through these numerical operations can introduce instability during training.[5] Moreover, since empirical second-moment matrices are estimated from mini-batch samples, the resulting gradients can be highly biased, potentially slowing convergence during optimization.

The VAMP-2 objective [48] suffers from similar issues, as it also requires computation of a matrix inside the Frobenius norm. The DPNet objective in Eq. (2) introduces the additional metric distortion loss $\mathcal{R}(\mathsf{M})$, which inherits both of the aforementioned issues. Similarly, the operator norms in the denominator of the DPNet-relaxed objective in Eq. (3) are subject to the same challenges.

## 3 Proposed Methods

In this section, we introduce an optimization framework based on the idea of *low-rank approximation* (LoRA), which circumvents the issues in the existing proposals. Although LoRA has been recently explored in various contexts [21, 45, 42, 39, 8, 32, 49, 17], its application to learning *parametric singular functions* of the Koopman operator remains largely unexplored. A detailed discussion of related formulations and prior work on LoRA is provided in Appendix A. After describing the learning procedure, we investigate two inference methods that build upon the learned representations.

### 3.1 Learning

To bypass the numerical and optimization challenges, we propose to directly minimize the low-rank approximation error $\|\mathcal{K} - \sum_{i=1}^{k} f_i \otimes g_i\|_{\mathrm{HS}}^2$ to find the Koopman singular functions. Succinctly, the learning objective, whose derivation is provided in Appendix C.1, can be expressed as

$$\mathcal{L}_{\mathsf{lora}}(\mathbf{f},\mathbf{g}) \triangleq -2\operatorname{tr}(\mathsf{T}[\mathbf{f},\mathbf{g}]) + \operatorname{tr}(\mathsf{M}_{\rho_0}[\mathbf{f}]\mathsf{M}_{\rho_1}[\mathbf{g}]). \tag{4}$$

The celebrated Eckart–Young–Mirsky theorem [6, 27] (or, more precisely, Schmidt's theorem [34]) establishes that this objective precisely characterizes the singular subspaces of $\mathcal{K}$ as a global optimum.

**Proposition 3.1** (Optimality of LoRA loss; see, e.g., [32, Theorem 3.1]). *Let $\mathcal{K}\colon L^2_{\rho_1}(\mathcal{X}) \to L^2_{\rho_0}(\mathcal{X})$ be a compact operator having SVD $\sum_{i=1}^{\infty} \sigma_i \phi_i \otimes \psi_i$ with $\sigma_1 \geq \sigma_2 \geq \ldots \geq 0$. Let $(\mathbf{f}^\star, \mathbf{g}^\star)$ denote a global minimizer of $\mathcal{L}_{\mathsf{lora}}(\mathbf{f},\mathbf{g})$. If $\sigma_k > \sigma_{k+1}$, then $\sum_{i=1}^{k} f_i \otimes g_i = \sum_{i=1}^{k} \sigma_i \phi_i \otimes \psi_i$.*

Moreover, owing to the simple form of the objective, its learning-theoretic properties are amenable to analysis. In particular, under a mild boundedness assumption, the empirical objective $\hat{\mathcal{L}}_{\mathsf{lora}}(\mathbf{f},\mathbf{g})$ converges to the population objective $\mathcal{L}_{\mathsf{lora}}(\mathbf{f},\mathbf{g})$ at the rate of $O_{\mathbb{P}}(N^{-1/2})$, when $N$ denotes the number of pairs from the trajectory data. We defer the statement of Theorem C.1 to Appendix C.2.

A notable property of the LoRA objective, compared to the VAMPnet and DPNet objectives, is that it is expressed entirely as a *polynomial* in the second-moment matrices. As a result, its gradient can be naturally estimated in an unbiased manner, in contrast to VAMPnet and DPNet, which require backpropagation through numerical linear-algebra operations. This property is particularly advantageous when optimizing large-scale models with moderately sized minibatches.

**Special Case: Reversible Continuous-Time Dynamics.** As argued in [16], we can also directly analyze reversible continuous-time dynamics, which includes an important example of (overdamped)

---

[5]In the PyTorch implementation, functions such as `lstsq`, `eigh`, and `matrix_norm` from the `torch.linalg` package are used; see, e.g., the PyTorch implementations of VAMPnet in `deeptime` and `kooplearn` libraries. For example, eigendecomposition (`torch.linalg.eigh`) may be numerically unstable when the matrices are ill-conditioned, since the gradients of eigenvectors scale inversely with eigenvalue gaps, which can lead to gradient explosions in nearly degenerate subspaces.

Langevin dynamics. In this special case, we can apply the spectral techniques under a weaker assumption than compactness; for example, we only require the largest eigenvalue to be separated from its essential spectrum; see, e.g., [12, Section III.4]. Our objective in Eq. (4) simplifies to

$$\mathcal{L}_{\mathsf{lora}}^{\mathsf{sa}}(\mathbf{f}) \triangleq -2\operatorname{tr}(\mathsf{M}_{\rho_0}[\mathbf{f}, \mathcal{L}\mathbf{f}]) + \|\mathsf{M}_{\rho_0}[\mathbf{f}]\|_{\mathrm{F}}^2, \tag{5}$$

where $\mathsf{M}_{\rho_0}[\mathbf{f}, \mathcal{L}\mathbf{f}] \triangleq \mathbb{E}[\mathbf{f}(\mathbf{x})(\mathcal{L}\mathbf{f})(\mathbf{x})^\intercal]$ is plugged in in place of $\mathsf{T}[\mathbf{f}, \mathbf{g}] = \mathsf{M}_{\rho_0}[\mathbf{f}, \mathcal{K}\mathbf{g}]$. Ryu et al. [32, Theorem C.5] show that the LoRA objective applied on a (possibly non-compact) self-adjoint operator can find the positive eigenvalues that are above its essential spectrum. We describe a special example of stochastic differential equations in Appendix G.3.

**Nesting Technique for Learning Ordered Singular Functions.** As introduced in [32], we can apply the *nesting* technique to directly learn the ordered singular functions. We note that learning the ordered singular functions is not an essential procedure, given that we only require well-learned singular subspaces during inference, as we explain below. We empirically found, however, that LoRA with nesting consistently improves overall downstream task performance. We conjecture that the nesting technique helps the parametric models to focus on the most important signals, and thus improves the overall convergence.

The key idea of nesting is to solve the LoRA problem for all dimensions $i \in [k]$ simultaneously. There are two versions proposed in [32], *joint* and *sequential* nesting. On one hand, in *joint nesting*, we simply aim to minimize a single objective $\mathcal{L}_{\mathsf{lora}}^{\mathsf{joint}}(\mathbf{f}, \mathbf{g}) \triangleq \sum_{i=1}^k \alpha_i \mathcal{L}_{\mathsf{lora}}(\mathbf{f}_{1:i}, \mathbf{g}_{1:i})$ for any choice of positive weights $\alpha_1, \ldots, \alpha_k > 0$, while the uniform weighting has been proven to be the most effective [32]. The joint objective characterizes the ordered singular functions as its unique global optima. On the other hand, *sequential nesting* iteratively update the $i$-th function pair $(f_i, g_i)$, using their gradient from $\mathcal{L}_{\mathsf{lora}}(\mathbf{f}_{1:i}, \mathbf{g}_{1:i})$, as if the previous modes $(\mathbf{f}_{1:i-1}, \mathbf{g}_{1:i-1})$ were perfectly fitted to the top-$(i-1)$ singular-subspaces. Given that different modes are independently parameterized, one can use an inductive argument to show the convergence of sequential nesting. Both joint and sequential nesting can be implemented efficiently, with almost no additional computational cost compared to the LoRA objective without nesting. We defer the details to Appendix F.

Ryu et al. [32] advocate using sequential nesting for separately parameterized networks, while recommending joint nesting otherwise. In all our experiments, however, we employed two neural networks $\mathbf{f}_\theta$ and $\mathbf{g}_\theta$ (i.e., assuming joint parameterization), and empirically observed that sequential nesting achieved convergence comparable to joint nesting. Thus, a practitioner may by default adopt joint nesting as a principled choice for jointly parameterized networks, while regarding sequential nesting as another option as a working heuristic. In the molecular dynamics experiment below, we found that the nesting technique can provide a stabilization effect during training; see Section 4.3 and Appendix G.4.

**Explicit Parameterization of Singular Values.** An alternative, yet natural parameterization is to explicitly parameterize the singular values by learnable parameters $\boldsymbol{\gamma} \in \mathbb{R}_{\geq 0}^k$, and plug in $\mathbf{f} \leftarrow \sqrt{\boldsymbol{\gamma}} \odot \mathbf{f}$ and $\mathbf{g} \leftarrow \sqrt{\boldsymbol{\gamma}} \odot \mathbf{g}$ to the LoRA objective in Eq. (4), under the unit-norm constraints $\|f_i\|_{\rho_0} = \|g_i\|_{\rho_1} = 1$ for all $i \in [k]$. Then, we get the explicit LoRA objective $\mathcal{L}_{\mathsf{lora}}^{\mathsf{explicit}}(\boldsymbol{\gamma}, \mathbf{f}, \mathbf{g}) \triangleq -2\operatorname{tr}(\mathsf{T}[\sqrt{\boldsymbol{\gamma}} \odot \mathbf{f}, \sqrt{\boldsymbol{\gamma}} \odot \mathbf{g}]) + \operatorname{tr}(\mathsf{M}_{\rho_0}[\sqrt{\boldsymbol{\gamma}} \odot \mathbf{f}] \mathsf{M}_{\rho_1}[\sqrt{\boldsymbol{\gamma}} \odot \mathbf{g}])$. In a similar spirit to [16], Kostic et al. [17] proposed a regularized objective $\mathcal{L}_{\mathsf{lora}}^{\mathsf{explicit}}(\boldsymbol{\gamma}, \mathbf{f}, \mathbf{g}) + \lambda(\mathcal{R}'_{\rho_0}[\mathbf{f}] + \mathcal{R}'_{\rho_1}[\mathbf{g}])$, where $\mathcal{R}'_{\rho_0}[\mathbf{f}] \triangleq \|\mathsf{M}_{\rho_0}[\mathbf{f}] - \mathsf{I}\|_{\mathrm{F}}^2 + 2\|\mathbb{E}_{\rho_0}[\mathbf{f}]\|^2$ and $\mathcal{R}'_{\rho_1}[\mathbf{g}]$ is similarly defined. An alternative implementation of the explicit parameterization involves $L_2$-batch normalization, as suggested by Deng et al. [5]. We experimented both formulations and empirically observed that the original parameterization in Eq. (4) performs well in practice, and no performance improvement was observed with these regularization approaches.

## 3.2 Inference

After fitting $\mathbf{f}(\cdot)$ and $\mathbf{g}(\cdot)$ to the top-$k$ singular subspaces, we can perform the downstream tasks such as (1) finding ordered singular functions, (2) performing eigen-analysis, and (3) multi-step prediction. We describe two approaches, each of which is a slightly extended version from VAMPnet and DPNet, respectively. The two approaches are based on rather different principles: Approach 1 (VAMPnet type) is based on estimating the Koopman operator by LoRA using both left and right singular functions, while Approach 2 (DPNet type) uses one set of basis and performs downstream tasks by projecting the Koopman operator onto the span of the basis. In the experiments below, we will compare and

Table 1: Summary of inference procedures given learned singular functions (or basis) $\mathbf{f}, \mathbf{g}$.

| | Approach 1. CCA + LoRA | Approach 2. EDMD ($\mathbf{b} \in \{\mathbf{f}, \mathbf{g}\}$) |
|---|---|---|
| CCA step 1: Whitening | $\tilde{\mathbf{f}}(\mathbf{x}) \triangleq (\mathsf{M}_{\rho_0}[\mathbf{f}])^{-1/2}\mathbf{f}(\mathbf{x})$ $\tilde{\mathbf{g}}(\mathbf{x}') \triangleq (\mathsf{M}_{\rho_1}[\mathbf{g}])^{-1/2}\mathbf{g}(\mathbf{x}')$ | N/A |
| CCA step 2: SVD | $\mathsf{T}[\tilde{\mathbf{f}}, \tilde{\mathbf{g}}] = \mathsf{U}\Sigma\mathsf{V}^\mathsf{T}$ | N/A |
| Ordered singular functions | $\tilde{\phi}(\mathbf{x}) = \Sigma^{1/2}\mathsf{U}^\mathsf{T}\tilde{\mathbf{f}}(\mathbf{x})$ $\tilde{\psi}(\mathbf{x}') = \Sigma^{1/2}\mathsf{V}^\mathsf{T}\tilde{\mathbf{g}}(\mathbf{x}')$ | N/A |
| Approximate Koopman matrix | $\mathsf{K}^{\text{right}}_{\tilde{\phi}, \tilde{\psi}} \triangleq \mathsf{M}_{\rho_1}[\tilde{\psi}, \tilde{\phi}]$ (right) $\mathsf{K}^{\text{left}}_{\tilde{\phi}, \tilde{\psi}} \triangleq \mathsf{M}_{\rho_0}[\tilde{\psi}, \tilde{\phi}]$ (left) | $\mathsf{K}^{\text{ols}}_{\mathbf{b}} \triangleq (\mathsf{M}_{\rho_0}[\mathbf{b}])^+\mathsf{T}[\mathbf{b}]$ (right) $\mathsf{K}^{+,\text{ols}}_{\mathbf{b}} \triangleq (\mathsf{M}_{\rho_1}[\mathbf{b}])^+\mathsf{T}[\mathbf{b}]$ (left) |
| Forward $\mathbb{E}_{p(\mathbf{x}_t \mid \mathbf{x}_0)}[h(\mathbf{x}_t)]$ | $\tilde{\phi}(\mathbf{x}_0)^\mathsf{T}(\mathsf{K}^{\text{right}}_{\tilde{\phi}, \tilde{\psi}})^{t-1}\langle h, \tilde{\psi}\rangle_{\rho_1}$ | $\mathbf{b}(\mathbf{x}_0)^\mathsf{T}(\mathsf{K}^{\text{ols}}_{\mathbf{b}})^t(\mathsf{M}_{\rho_0}[\mathbf{b}])^+\langle h, \mathbf{b}\rangle_{\rho_0}$ |
| Backward $\mathbb{E}_{p(\mathbf{x}_0 \mid \mathbf{x}_t)}[h(\mathbf{x}_0)]$ | $\tilde{\psi}(\mathbf{x}_t)^\mathsf{T}(\mathsf{K}^{\text{left},\mathsf{T}}_{\tilde{\phi}, \tilde{\psi}})^{t-1}\langle h, \tilde{\phi}\rangle_{\rho_0}$ | $\mathbf{b}(\mathbf{x}_0)^\mathsf{T}(\mathsf{K}^{+,\text{ols}}_{\mathbf{b}})^t(\mathsf{M}_{\rho_1}[\mathbf{b}])^+\langle h, \mathbf{b}\rangle_{\rho_1}$ |

discuss their pros and cons. For simplicity, we describe the procedures using population quantities; in practice, they are replaced with empirical estimates (i.e., $\mathsf{M} \leftarrow \hat{\mathsf{M}}$ and $\mathsf{T} \leftarrow \hat{\mathsf{T}}$).

### 3.2.1 Approach 1: CCA + LoRA

In a similar spirit to the inference procedure of VAMPnet [48], we can perform a canonical correlation analysis (CCA) [10] to retrieve the (ordered) singular values and singular functions as follows, given that $\mathbf{f}(\cdot)$ and $\mathbf{g}(\cdot)$ capture left and right singular subspaces. First, we define the *whitened* basis functions $\tilde{\mathbf{f}}(\mathbf{x}) \triangleq (\mathsf{M}_{\rho_0}[\mathbf{f}])^{-1/2}\mathbf{f}(\mathbf{x})$ and $\tilde{\mathbf{g}}(\mathbf{x}') \triangleq (\mathsf{M}_{\rho_1}[\mathbf{g}])^{-1/2}\mathbf{g}(\mathbf{x}')$. Then, we perform the SVD of the joint second moment matrix $\mathsf{T}[\tilde{\mathbf{f}}, \tilde{\mathbf{g}}] = \mathsf{U}\Sigma\mathsf{V}^\mathsf{T}$, where $\mathsf{U} \in \mathbb{R}^{k \times r}, \Sigma \in \mathbb{R}^{r \times r}, \mathsf{V} \in \mathbb{R}^{k \times r}$. We define *aligned* singular functions as

$$\tilde{\phi}(\mathbf{x}) \triangleq \Sigma^{1/2}\mathsf{U}^\mathsf{T}\tilde{\mathbf{f}}(\mathbf{x}) \in \mathbb{R}^r \quad \text{and} \quad \tilde{\psi}(\mathbf{x}') \triangleq \Sigma^{1/2}\mathsf{V}^\mathsf{T}\tilde{\mathbf{g}}(\mathbf{x}) \in \mathbb{R}^r, \tag{6}$$

and approximate the transition kernel as $k(\mathbf{x}, \mathbf{x}') = \frac{p(\mathbf{x}' \mid \mathbf{x})}{\rho_1(\mathbf{x}')} \approx \tilde{\phi}(\mathbf{x})^\mathsf{T}\tilde{\psi}(\mathbf{x}') = \tilde{\mathbf{f}}(\mathbf{x})^\mathsf{T}\mathsf{T}[\tilde{\mathbf{f}}, \tilde{\mathbf{g}}]\tilde{\mathbf{g}}(\mathbf{x}')$. Hence, compared to the direct approximation $\mathbf{f}(\mathbf{x})^\mathsf{T}\mathbf{g}(\mathbf{x}')$, the CCA procedure *whitens* (by $\mathsf{M}_{\rho_0}[\mathbf{f}]$ and $\mathsf{M}_{\rho_1}[\mathbf{g}]$) and *corrects* (by SVD of $\mathsf{T}[\tilde{\mathbf{f}}, \tilde{\mathbf{g}}]$) the given basis. In a real-world scenario, the CCA alignment can always help $\mathbf{f}, \mathbf{g}$ better aligned, and improve the quality of singular value estimation.

Given this finite-rank approximation, the eigenfunctions can be reconstructed using the following approximate Koopman matrices, based on the theory developed for finite-rank Koopman operators in Appendix D. Specifically, the eigenvalues of $\mathcal{K}$ and the right eigenfunctions can be approximated using the matrix $\mathsf{K}^{\text{right}}_{\tilde{\phi}, \tilde{\psi}} \triangleq \mathsf{M}_{\rho_1}[\tilde{\psi}, \tilde{\phi}]$; see Theorem D.2. Similarly, the left eigenfunctions and corresponding eigenvalues can be approximated using $\mathsf{K}^{\text{left}}_{\tilde{\phi}, \tilde{\psi}} \triangleq \mathsf{M}_{\rho_0}[\tilde{\psi}, \tilde{\phi}]$; see Theorem D.3. We note that this eigen-analysis using the finite-rank approximation is new compared to [48].

Lastly, given the LoRA $\frac{p(\mathbf{x}' \mid \mathbf{x})}{\rho_1(\mathbf{x}')} \approx \tilde{\phi}(\mathbf{x})^\mathsf{T}\tilde{\psi}(\mathbf{x}')$, the conditional expectation can be approximated as

$$\mathbb{E}_{p(\mathbf{x}' \mid \mathbf{x})}[h(\mathbf{x}')] = \mathbb{E}_{\rho_1(\mathbf{x}')}\left[\frac{p(\mathbf{x}' \mid \mathbf{x})}{\rho_1(\mathbf{x}')}h(\mathbf{x}')\right] \approx \sum_{i=1}^{k}\tilde{\phi}_i(\mathbf{x})\langle\tilde{\psi}_i, h\rangle_{\rho_1} = \tilde{\phi}(\mathbf{x})^\mathsf{T}\langle h, \tilde{\psi}\rangle_{\rho_1},$$

where we let $\langle h, \tilde{\psi}\rangle_{\rho_1} \triangleq \left[\langle h, \tilde{\psi}_1\rangle_{\rho_1}, \dots, \langle h, \tilde{\psi}_k\rangle_{\rho_1}\right]^\mathsf{T} \in \mathbb{R}^k$. Extending the reasoning, we can obtain an approximate multi-step prediction as

$$\mathbb{E}_{p(\mathbf{x}_t \mid \mathbf{x}_0)}[h(\mathbf{x}_t)] \approx \tilde{\phi}(\mathbf{x}_0)^\mathsf{T}(\mathsf{K}^{\text{right}}_{\tilde{\phi}, \tilde{\psi}})^{t-1}\langle h, \tilde{\psi}\rangle_{\rho_1}$$
$$= \tilde{\mathbf{f}}(\mathbf{x}_0)^\mathsf{T}\mathsf{T}[\tilde{\mathbf{f}}, \tilde{\mathbf{g}}](\mathsf{M}_{\rho_1}[\tilde{\mathbf{g}}, \tilde{\mathbf{f}}]\mathsf{T}[\tilde{\mathbf{f}}, \tilde{\mathbf{g}}])^{t-1}\langle h, \tilde{\mathbf{g}}\rangle_{\rho_1}. \tag{7}$$

If $\mathsf{M}_{\rho_1}[\tilde{\mathbf{g}}, \tilde{\mathbf{f}}]\mathsf{T}[\tilde{\mathbf{f}}, \tilde{\mathbf{g}}]$ is diagonalizable, we can perform its EVD to make the matrix power computation more efficient. Similarly, we can perform the multi-step *backward* prediction using $\mathsf{K}^{\text{left}}_{\tilde{\phi}, \tilde{\psi}}$ as follows:

$$\mathbb{E}_{p(\mathbf{x}_0 \mid \mathbf{x}_t)}[h(\mathbf{x}_0)] \approx \tilde{\mathbf{g}}(\mathbf{x}_0)^\mathsf{T}\mathsf{T}[\tilde{\mathbf{g}}, \tilde{\mathbf{f}}](\mathsf{M}_{\rho_0}[\tilde{\mathbf{f}}, \tilde{\mathbf{g}}]\mathsf{T}[\tilde{\mathbf{g}}, \tilde{\mathbf{f}}])^{t-1}\langle h, \tilde{\mathbf{f}}\rangle_{\rho_0}. \tag{8}$$

### 3.2.2 Approach 2: Extended DMD

Once a good subspace $\mathsf{span}(\mathbf{b})$ has been learned using some basis function $\mathbf{b}\colon \mathcal{X} \to \mathbb{R}^k$, Kostic et al. [16] proposed to perform the *operator regression* [14] (also known as *principal component regression* [15]), which is essentially the EDMD [46]. The EDMD approximates the Koopman operator by $\mathsf{K}_{\mathbf{b}}^{\mathsf{ols}} \triangleq (\mathsf{M}_{\rho_0}[\mathbf{b}])^+ \mathsf{T}[\mathbf{b}] \in \mathbb{R}^{k \times k}$, which can be understood as the best finite-dimensional approximation of the Koopman operator $\mathcal{K}$ restricted on $\mathsf{span}(\mathbf{b})$, in the sense that $(\mathcal{K}\mathbf{b})(\mathbf{x}) = \mathbb{E}_{p(\mathbf{x}' \mid \mathbf{x})}[\mathbf{b}(\mathbf{x}')] \approx \mathsf{K}\mathbf{b}(\mathbf{x})$. Now, given a function $h \in \mathsf{span}(\mathbf{b})$, we can again choose the least-squares solution $\mathbf{z}_{\mathbf{b}}^{\mathsf{ols}}(h) \triangleq (\mathsf{M}_{\rho_0}[\mathbf{b}])^+ \langle h, \mathbf{b} \rangle_{\rho_0}$ to find the best $\mathbf{z}$ such that $h(\mathbf{x}) \approx \mathbf{z}^{\intercal}\mathbf{b}(\mathbf{x})$ for $\mathbf{x} \sim \rho_0(\mathbf{x})$. Given this, we can finally approximate the multi-step prediction as

$$\mathbb{E}_{p(\mathbf{x}_t \mid \mathbf{x}_0)}[h(\mathbf{x}_t)] \approx \mathbf{b}(\mathbf{x}_0)^{\intercal}(\mathsf{K}_{\mathbf{b}}^{\mathsf{ols}})^t \mathbf{z}_{\mathbf{b}}^{\mathsf{ols}}(h) = \mathbf{b}(\mathbf{x}_0)^{\intercal}((\mathsf{M}_{\rho_0}[\mathbf{b}])^+ \mathsf{T}[\mathbf{b}])^t (\mathsf{M}_{\rho_0}[\mathbf{b}])^+ \langle h, \mathbf{b} \rangle_{\rho_0}. \quad (9)$$

Applying the same logic to the adjoint operator $\mathcal{K}^*$, we can perform the backward prediction as

$$\mathbb{E}_{p(\mathbf{x}_0 \mid \mathbf{x}_t)}[h(\mathbf{x}_0)] \approx \mathbf{b}(\mathbf{x}_t)^{\intercal}((\mathsf{M}_{\rho_1}[\mathbf{b}])^+ \mathsf{T}[\mathbf{b}]^{\intercal})^t (\mathsf{M}_{\rho_1}[\mathbf{b}])^+ \langle h, \mathbf{b} \rangle_{\rho_1}. \quad (10)$$

We defer a more detailed derivation of the EDMD predictions to Appendix E. We note that Kostic et al. [16] proposed to use the *left* singular basis $\mathbf{f}$ for the basis $\mathbf{b}$, whereas we empirically found that using the *right* singular basis $\mathbf{g}$ yields comparable results.

## 4 Experiments

We demonstrate the efficacy of the proposed techniques using the experimental suite of [16]. Unless stated otherwise, all experimental settings are mostly identical to those in [16] except the molecular dynamics experiment. All technical details and configurations are provided in Appendix G. The appendix also includes an additional experiment on an instance of a 1D noisy logistic map, whose Koopman operator has finite rank. We defer this result to the appendix, as most methods perform comparably in this simple setting. Our PyTorch [30] implementation is publicly available at `https://github.com/MinchanJeong/NeuralKoopmanSVD`.

### 4.1 Ordered MNIST

We considered the ordered MNIST example, a synthetic experiment which was first considered in [14]: given an MNIST image $\mathbf{x}_t$ with digit $y_t \in \{0, \dots, 4\}$, $\mathbf{x}_{t+1}$ is drawn at random from the MNIST images of digit $y_{t+1} = y_t + 1 \pmod 5$. While this process is not time-reversible, the process is clearly normal. We tested VAMPnet-1, DPNet, DPNet-relaxed, and sequentially and jointly nested version of LoRA. For each method, we trained singular basis parameterized by convolutional neural networks with 100 epochs using 10 random seeds.

We evaluated the multi-step prediction performance of each method, by computing (1) the accuracy using an oracle classifier similar to [16], and (2) root mean squared error (RMSE) of the prediction evaluated on the test data. The multi-step prediction performed with EDMD($\mathbf{g}$) is reported in the first row of Figure 1. We highlight that LoRA and its variants consistently outperform the other methods, exhibiting robust RMSE performance over a range of prediction steps $t \in \{-15, \dots, 15\}$. Notably, the nesting techniques helped improve the RMSE, especially the joint nesting worked best. We also remark in passing that, unlike the significant failure reported in [16], the VAMPnet-1 prediction quality is comparable to DPNet in both metrics.

In the second row, we showed the performance of different prediction methods with the basis learned with LoRA$_{\mathsf{jnt}}$. We note that the quality is very close to each other, and CCA+LoRA prediction seems to follow the trend of EDMD($\mathbf{f}$) for forward prediction and EDMD($\mathbf{g}$) for backward prediction. We also remark that the prediction quality with CCA+LoRA when $t = 1$ is particularly bad, which we conjecture to be caused by the absence of the action of Koopman matrices; see Eq. (7) and Eq. (8).

### 4.2 Langevin Dynamics

We also tested the performance of LoRA$_{\mathsf{seq}}$ with the objective in Eq. (5) to learn the eigenfunctions of a 1D Langevin dynamics, which is a continuous-time, time-reversible process; see Appendix G.3 for the stochastic differential equation used in the experiment.

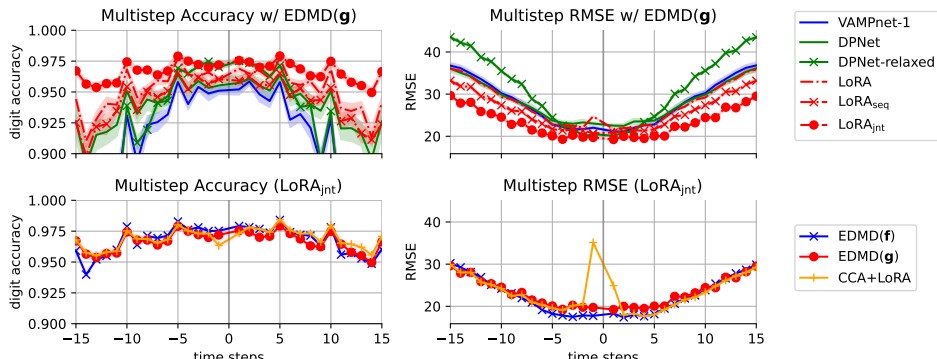

Figure 1: Summary of the MNIST experiment. The shaded area indicates $\pm 1$ standard deviation.

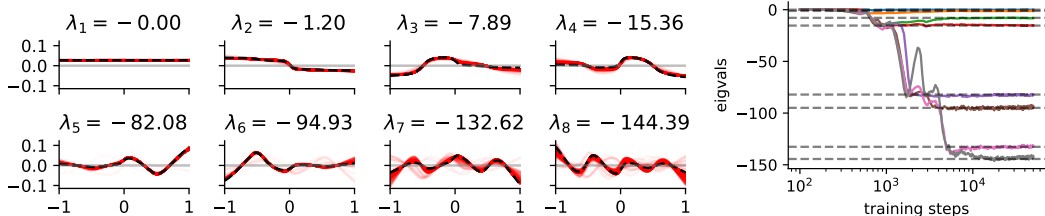

Figure 2: Visualization of the eigenfunctions of the 1D Langevin dynamics, learned by $\text{LoRA}_{\text{seq}}$. In the first panel, the learned eigenfunctions across training iterations are overlaid, with later iterations displayed with higher opacity (red). The dashed lines indicate the true eigenfunctions.

We drew a sample trajectory of length $7 \times 10^4$ by the Euler–Maruyama scheme and used it for training. We parameterized the top-10 eigenfunctions using a single MLP with hidden units `128-128-128` and CeLU activation. We trained the network using Adam optimizer with learning rate $10^{-3}$ for 50,000 iterations with batch size 128. The differentiation operation in the generator was computed by the autodiff feature of PyTorch. Exponential moving average with decay 0.995 was applied to result in a smoother result. In Figure 2, we report the first eight eigenfunctions and the convergence behavior of the estimated eigenvalues, without any postprocessing other than sign alignment. As demonstrated, the learned eigenfunctions and eigenvalues converge to the ground truths reliably with $\text{LoRA}_{\text{seq}}$.

We also implemented the DPNet objective $\tilde{\mathcal{L}}_{\text{dp}}^{(\gamma)}(\mathbf{f}) \triangleq -\operatorname{tr}(\mathsf{M}_{\rho_0}[\mathbf{f}]^{\dagger}\mathsf{M}_{\rho_0}[\mathbf{f}, \mathcal{L}\mathbf{f}]) + \gamma\mathcal{R}(\mathsf{M}_{\rho_0}[\mathbf{f}])$, following the recommended configuration in [16], but we were unable to obtain successful results. This example demonstrates the effectiveness of the LoRA framework for a continuous-time dynamics.

### 4.3 Chignolin Molecular Dynamics

To assess scalability on high-dimensional data, we apply our method to the analysis of *chignolin* molecular dynamics. Chignolin is an artificial mini-protein that is considered a standard model for studying rapid folding dynamics due to its complex and fast transitions [20, 3, 16]. The underlying physical system is time-reversible, and thus the Koopman operator is self-adjoint. The slowest decay mode in chignolin is associated with the folding-unfolding transition, which occurs on a microsecond timescale [20]. Unlike [16], we use the public dataset of [25]. To isolate algorithmic stability from variance due to data sampling, we fix a data split chosen to balance the root mean square distance statistics across splits. All experimental details are provided in Appendix G.4.

We first remark on the overall quality of the learned models. As reported in [16], we observe that VAMPnet-2 and DPNet diverge. However, we empirically find that models trained with VAMPnet-1 can converge, albeit with several caveats. First, we observe that VAMPnet-1 is sensitive to low-level numerical implementations in `torch.linalg`, and diverges under a specific PyTorch version. Moreover, we observe that VAMPnet-1 diverges for certain alternative train-test splits that are not reported here. Consequently, the only methods that consistently exhibit convergence in our experiments are our LoRA variants. We nonetheless report VAMPnet-1 results to demonstrate that our method outperforms it even in regimes where VAMPnet-1 converges.

Table 2: Test VAMP-E scores on the 300 K chignolin dataset [25] reported as the mean $\pm$ 95% confidence interval (CI). The low-rank case reports results over 5 random seeds. In the high-rank case, CI is calculated with the final 5 checkpoints of a single run. (*Note:* We omit the numerically unstable baselines VAMPnet-2 and DPNet.)

| | Low-rank ($k = 16$) | High-rank ($k = 64$) | |
| Algorithm | $H = 64, B = 384$ | $H = 128, B = 384$ | $H = 64, B = 96$ |
| --- | --- | --- | --- |
| DPNet-relaxed | $7.36_{\pm 0.40}$ | $7.85_{\pm 0.24}$ | $6.97_{\pm 0.31}$ |
| VAMPnet-1 | $9.54_{\pm 0.31}$ | $34.76_{\pm 0.50}$ | $19.71_{\pm 0.59}$ |
| LoRA | $10.27_{\pm 0.31}$ | $38.89_{\pm 0.29}$ | $37.74_{\pm 0.95}$ |
| LoRA$_{\text{jnt}}$ | $10.74_{\pm 0.35}$ | $39.37_{\pm 0.24}$ | $\mathbf{38.50}_{\pm 0.83}$ |
| LoRA$_{\text{seq}}$ | $\mathbf{12.29}_{\pm 0.07}$ | $\mathbf{42.44}_{\pm 0.35}$ | $37.33_{\pm 1.66}$ |

**Operator Approximation Performance.** Table 2 reports the VAMP-E scores which measure the operator approximation error in the Hilbert–Schmidt norm. In the low-rank case with $k = 16$, LoRA variants consistently outperform baselines, with LoRA$_{\text{seq}}$ achieving the highest score and lowest variance. We further assessed stability under a high-rank setting with $k = 64$. As shown in the right columns of Table 2, VAMPnet-1 exhibits significant performance degradation in the restricted-batch setting. We attribute this behavior to numerical instability arising from ill-conditioned moment matrices when using small batches. In contrast, our LoRA variants maintain strong performance across all regimes, demonstrating superior robustness.

**Test-Time Orthogonality Analysis.** As another performance measure, we report the test-time orthogonality of the learned basis. To this end, we compute the Gram matrix of the whitened basis with respect to the test data, formally calculated as $\mathsf{M}_{\rho_0^{\text{train}}}[\mathbf{f}]^{-\frac{1}{2}} \mathsf{M}_{\rho_0^{\text{test}}}[\mathbf{f}] \mathsf{M}_{\rho_0^{\text{train}}}[\mathbf{f}]^{-\frac{1}{2}}$, and similarly for $\mathbf{g}$. Ideally, this quantity should be an identity matrix.

As illustrated in Figure 3, LoRA variants maintain a relatively cleaner diagonal structure than the baselines, which exhibit sensitivity to distribution shifts. This confirms that the bases learned by LoRA more effectively capture the invariant geometry of the system.

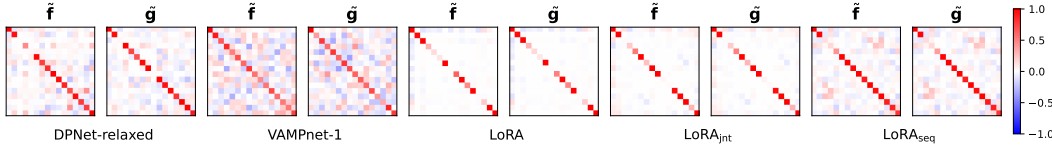

Figure 3: Test-set orthogonality evaluation of the learned bases whitened by training data. LoRA variants preserve a clear diagonal structure, demonstrating superior generalization.

# 5 Concluding Remarks

The evolution of linear-algebraic tools for Koopman analysis, such as DMD, extended DMD, kernel DMD, VAMPnet, and DPNet, mirrors that of CCA and its kernel and maximal-correlation variants. The LoRA framework for Koopman operator can be similarly understood as an adoption of the recent advances in correlation analysis and representation learning, which can naturally avoid unstable decompositions and train efficiently with standard mini-batch gradients. Across diverse benchmarks, LoRA variants deliver superior singular subspaces, eigenfunctions, long-horizon forecasts, and scalability, establishing a practical, reliable path for data-driven modeling of complex dynamics.

While the LoRA framework represents an important step toward scalable deep learning methods for analyzing stochastic dynamical systems, key limitations persist. For example, as shown in the chignolin experiment, the quality of learned dynamics is inherently constrained by the data's temporal resolution, which can prevent the recovery of the slowest physical processes. Furthermore, robustly identifying coherent structures in highly non-normal or chaotic systems remains an open challenge.

Future research should therefore focus on advancing the deep learning methodologies tailored for Koopman analysis. This includes developing specialized neural architectures and establishing stronger theoretical foundations for learning from trajectories of dynamical systems. A deeper understanding of the landscape of the LoRA objective will also be crucial for advancing this line of work.

## Acknowledgments

JJR and GWW were supported in part by the MIT-IBM Watson AI Lab under Agreement No. W1771646. MJ and SYY were supported by the Institute for Information & Communications Technology Planning & Evaluation (IITP) grant funded by the Korean government (MSIT) [No. RS-2022-II220311, Development of Goal-Oriented Reinforcement Learning Techniques for Contact-Rich Robotic Manipulation of Everyday Objects, 80%], [No. RS-2024-00457882, AI Research Hub Project, 10%], and [No. RS-2019-II190075, Artificial Intelligence Graduate School Program (KAIST), 10%].

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

# Appendix

# A  Related Work

In this section, we discuss some related work to better contextualize our contribution.

## A.1  Deep Learning Approaches for Koopman Operator Estimation

Deep learning models are increasingly being integrated into the Koopman theory framework to enable the learning of complex system dynamics by leveraging the expressive power of neural networks. One of the earliest examples is Koopman Autoencoders (KAEs) for forecasting sequential data [47, 37, 23, 2]. The KAE approach in general aims to learn a latent embedding and a global, time-invariant linear operator, guided by reconstruction and latent-space constraints. While these methods laid important groundwork, more recent developments [24, 15, 16] have demonstrated superior performance. Consequently, our experiments focus on these more advanced techniques.

Recently, various deep learning architectures for learning Koopman operator have been proposed to handle complex real-world environments, such as non-stationarity in time series data. For example, Liu et al. [22] introduce Fourier filters for disentanglement and distinct Koopman predictors for time-variant and invariant components. Another example is Koopman Neural Forecaster [43], which utilizes predefined measurement functions, a transformer-based local operator, and a feedback loop. We remark, however, that while these methods demonstrate impressive empirical results on some datasets, they mainly prioritize forecasting accuracy rather than eigen-analysis essential for understanding the underlying system and achieving robust prediction, which lies at the core of the Koopman framework. In this work, we focus on this core aspect, leaving forecasting extensions for future study.

## A.2 On Low-Rank Approximation

The idea of LoRA as a variational characterization for SVD of a matrix or operator has a century-long history; see [36] for an overview. Despite its simple and elegant formulation, dominated by the common Rayleigh quotient maximization framework and its variants, its wide adoption as a variational framework for computing singular subspaces based on optimization started only recently. For example, see [21, 45] from the numerical optimization literature, [8, 44] in representation learning, and [42, 49, 17] for correlation analysis. We refer the reader to [32, Appendix B] for a more detailed overview of the literature.

Our learning technique closely follows the low-rank-approximation-based framework dubbed as *NeuralSVD* in [32], including the nesting techniques. Ryu et al. [32] proposed the LoRA framework with nesting as a generic tool to perform SVD of a general compact operator using neural networks including scientific simulation and representation learning. While they also considered decomposing the density ratio $\frac{p(\mathbf{y}\,|\,\mathbf{x})}{p(\mathbf{y})}$, which they call the *canonical dependence kernel* (CDK), as a special case, they did not explicitly consider its application to dynamical systems. We note in passing that there exists a similar squared loss studied in density ratio estimation [11, 39], but it becomes only related to spectral decomposition when the underlying density ratio is in the form of CDK $\frac{p(x,y)}{p(x)p(y)}$.

More recently, in a concurrent work, Turri et al. [41] also proposed to use the LoRA loss, emphasizing a self-supervised learning perspective on learning singular functions. Specifically, they parameterize the encoder-lagged-encoder pair by defining the latter as $\mathbf{g}(\mathbf{x}') \triangleq \mathsf{P}\mathbf{f}(\mathbf{x}')$, where $\mathsf{P} \in \mathbb{R}^{k \times k}$ is a learnable matrix and $\mathbf{f}(\cdot)$ is a neural network encoder. For a fixed encoder $\mathbf{f}(\cdot)$, they show that the optimal $\mathsf{P}$ under the LoRA objective coincides with a least-squares estimator for the CDK. A systematic study of the benefits of this parameterization is an interesting direction for future work.

During the course of our independent study, we also identified an unpublished objective named `EYMLoss` in the GitHub repository of Kostic et al. [16], which effectively corresponds to the LoRA loss investigated in this work; see `https://github.com/Machine-Learning-Dynamical-Systems/kooplearn/blob/ca71864469576b39621e4d4e93c0439682166d1e/kooplearn/nn/losses.py#L79C7-L79C14`. The associated commit message indicates that the authors were unable to obtain satisfactory results with this formulation. We attribute this failure to a subtle but critical oversight in handling the constant modes. Recall that a Koopman operator, as a special case of CDK operators, admits constant functions as its leading singular functions. In our implementation, we explicitly account for this structure by prepending constant components (i.e., appending 1's) to the outputs of both the encoder and the lagged encoder. This ensures that the constant modes are appropriately represented and preserved throughout training and inference. In contrast, `EYMLoss` incorporates the constant modes by projecting out their contribution from the observables. Specifically, it replaces $\mathbf{f}(\mathbf{x})$ with $\mathbf{f}(\mathbf{x}) - \mathbb{E}_{\rho_0(\mathbf{x})}[\mathbf{f}(\mathbf{x})]$ in the LoRA loss, where the expectation is approximated empirically using minibatch samples. This approach implicitly enforces orthogonality to constant modes by requiring the learned singular functions to be zero-mean with respect to the data distribution. While this treatment is valid in principle, it necessitates that the encoded features remain centered not only during training but also at inference time. Moreover, the constant singular functions must be explicitly included in the final model representation. These additional steps do not appear to have been fully implemented in the `kooplearn` codebase, which may have contributed to the method's lack of empirical success.

Finally, we remark in passing that a recent work studies another unconstrained variational optimization framework, called the *orbital minimization method*, for decomposing positive definite (PD) operators using neural networks [33]. As the framework is restricted to PD operators, it is not directly applicable to the setup considered in the present manuscript. Developing an extension of this method for analyzing stochastic dynamical systems would be of theoretical interest.

## A.3 On Nesting Techniques

The sequential nesting technique proposed in [32] can be understood as a function-space version of the *triangularized orthogonalization-free method* studied in [7]. The joint nesting technique was first proposed specifically for CDK by Xu and Zheng [49], and later extended as a generic tool in Ryu et al. [32]. For a more discussion on the history of nesting, we refer to [32].

# B  Variational Principles for VAMPnet and DPNet

In this section, we review the principles behind VAMPnet and DPNet in more details.

## B.1  VAMPnet

VAMPnet [48, 24] is based on the following trace-maximization-type characterization of the top-$k$ singular subspaces.

**Theorem B.1.** *Let* $\mathcal{T}\colon L^2_{\rho_y}(\mathcal{Y}) \rightarrow L^2_{\rho_x}(\mathcal{X})$ *be a compact operator with singular triplets* $\{(\sigma_i, \phi_i(\cdot), \psi_i(\cdot))\}_{i=1}^{\infty}$, *i.e.,* $(\mathcal{T}\psi_i)(x) = \sigma_i\phi_i(x)$, $(\mathcal{T}^*\phi_i)(y) = \sigma_i\psi_i(y)$, *and* $\sigma_1 \geq \sigma_2 \geq \ldots \geq 0$. *Then, for any fixed* $r = 1, 2, \ldots$, *the optimizers of the following optimization problem*

$$\max_{\{(f_i, g_i)\}_{i=1}^k \subset L^2_{\rho_x}(\mathcal{X}) \times L^2_{\rho_y}(\mathcal{Y})} \quad \sum_{i=1}^k \langle f_i, \mathcal{T}g_i \rangle^r_{\rho_x}$$
$$\textit{subject to} \quad \mathsf{M}_{\rho_x}[\mathbf{f}] = \mathsf{M}_{\rho_y}[\mathbf{g}] = \mathsf{I}_k \tag{11}$$

*characterize the top-$k$ singular functions of* $\mathcal{T}$ *up to an orthogonal transformation.*

We refer an interested reader to the proof in [48]. For the particular case of $\mathcal{T}$ being a Koopman operator $\mathcal{K}\colon L^2_{\rho_1}(\mathcal{X}) \to L^2_{\rho_0}(\mathcal{X})$, Wu and Noé [48] named the objective

$$\mathcal{R}_r[\mathbf{f}, \mathbf{g}] \triangleq \sum_{i=1}^k \langle f_i, \mathcal{K}g_i \rangle^r_{\rho_0} \tag{12}$$

as the *VAMP-r score* of $\mathbf{f}$ and $\mathbf{g}$. It is noted that VAMP-1 score is identical to the objective of deep CCA [1]. Wu and Noé [48], Mardt et al. [24] advocated the use of VAMP-2 score, based on its relation to the $L_2$-approximation error.

In the original VAMP framework [48, 24], neural networks are not directly trained by the VAMP-$r$ score. Instead, they considered a two-stage procedure as follows:

- **Basis learning**: They parameterize the singular functions as

$$\mathbf{f}_{\mathsf{U},\theta}(\mathbf{x}) \triangleq \mathsf{U}^\mathsf{T}\mathbf{f}_\theta(\mathbf{x}),$$
$$\mathbf{g}_{\mathsf{V},\theta}(\mathbf{x}') \triangleq \mathsf{V}^\mathsf{T}\mathbf{g}_\theta(\mathbf{x}'),$$

  where $\mathbf{f}_\theta(\mathbf{x}) \triangleq (f_{\theta,1}(\mathbf{x}), \ldots, f_{\theta,k_0}(\mathbf{x}))^\mathsf{T} \in \mathbb{R}^{k_0}$ and $\mathbf{g}_\theta(\mathbf{x}') \triangleq (g_{\theta,1}(\mathbf{x}'), \ldots, g_{\theta,k_1}(\mathbf{x}'))^\mathsf{T} \in \mathbb{R}^{k_1}$ denote some basis functions, which in the trainable basis case is typically parameterized by neural networks, and $\mathsf{U} \in \mathbb{R}^{k_0 \times k}$ and $\mathsf{V} \in \mathbb{R}^{k_1 \times k}$ denote the projection matrix to further express the singular functions using the given basis. Since the jointly optimizing over all $\theta, \mathsf{U}, \mathsf{V}$ is not practical, to train $\mathbf{f}_\theta$ and $\mathbf{g}_\theta$, they plug-in $\mathbf{f}_{\mathsf{U},\theta}(\mathbf{x})$ and $\mathbf{g}_{\mathsf{V},\theta}(\mathbf{x}')$ into Eq. (11), and consider the best $\mathsf{U}$ and $\mathsf{V}$ by solving the partial maximization problem, i.e.,

$$\max_{\mathsf{U},\mathsf{V}} \mathcal{R}_r(\boldsymbol{\phi}, \boldsymbol{\psi}) = \left\| (\mathsf{M}_{\rho_0}[\mathbf{f}])^{-\frac{1}{2}} \mathsf{T}[\mathbf{f}, \mathbf{g}] (\mathsf{M}_{\rho_1}[\mathbf{g}])^{-\frac{1}{2}} \right\|_r^r.$$

  Here, $\|\cdot\|_r$ denotes the Schatten $r$-norm. This objective is what we call the (maximal) VAMP-$r$ score in our paper, which is the objective in the VAMP framework that trains the trainable basis functions.

- **Inference**: Once $\mathbf{f}_\theta$ and $\mathbf{g}_\theta$ are trained, Eq. (11) become an optimization problem over $\mathsf{U}$ and $\mathsf{V}$:

$$\max_{\mathsf{U},\mathsf{V}} \quad \mathcal{R}_r(\mathsf{U}, \mathsf{V})$$
$$\textit{subject to} \quad \mathsf{U}^\mathsf{T}\mathsf{M}_{\rho_0}[\mathbf{f}]\mathsf{U} = I$$
$$\mathsf{V}^\mathsf{T}\mathsf{M}_{\rho_1}[\mathbf{g}]\mathsf{V} = I,$$

  where

$$\mathcal{R}_r(\mathsf{U}, \mathsf{V}) \triangleq \sum_{i=1}^k (\mathbf{u}_i^\mathsf{T}\mathsf{T}[\mathbf{f}, \mathbf{g}]\mathbf{v}_i)^r.$$

Wu and Noé [48] proposed to solve this using the (linear) CCA algorithm [10], and calling the final algorithm *feature TCCA*. To avoid the degeneracy, basis functions need to be whitened. Strictly speaking, however, the feature TCCA algorithm has nothing to do with the optimization problem. We note that our CCA+LoRA approach is built upon this inference method in the VAMP framework.

Let $\hat{\sigma}_1, \ldots, \hat{\sigma}_k$ denote the top-$k$ singular values of the approximate *Koopman matrix* $\hat{\mathsf{K}} \triangleq (\mathsf{M}_{\rho_0}[\mathbf{f}])^{-\frac{1}{2}} \mathsf{T}[\mathbf{f}, \mathbf{g}](\mathsf{M}_{\rho_1}[\mathbf{g}])^{-\frac{1}{2}}$. After CCA, we can consider the rank-$k$ approximation of the underlying Koopman operator $\mathcal{K}$ as we considered in our Approach 1 in Section 3.2. If we call the $\hat{\mathcal{K}}$ denote the corresponding approximate operator, Wu and Noé [48] called the (shifted and negated) approximation error in the Hilbert–Schmidt norm

$$\mathcal{R}_{\mathrm{E}} \triangleq \|\mathcal{K}\|_{\mathrm{HS}}^2 - \|\hat{\mathcal{K}} - \mathcal{K}\|_{\mathrm{HS}}^2,$$

which is called the *VAMP-E score*. While this looks essentially identical to the low-rank approximation error we consider in this paper, Wu and Noé [48] and Mardt et al. [24] used this metric only for evaluation, but not considered for training. In Appendix G.4, we provide a precise definition of the VAMP-E score for completeness.

## B.2 DPNet

As introduced in the main text, the DPNet objective is given as

$$\mathcal{L}_{\mathrm{dp}}^{(\gamma)}(\mathbf{f}, \mathbf{g}) \triangleq -\left\| (\mathsf{M}_{\rho_0}[\mathbf{f}])^{-\frac{1}{2}} \mathsf{T}[\mathbf{f}, \mathbf{g}](\mathsf{M}_{\rho_1}[\mathbf{g}])^{-\frac{1}{2}} \right\|_{\mathrm{F}}^2 + \gamma(\mathcal{R}(\mathsf{M}_{\rho_0}[\mathbf{f}]) + \mathcal{R}(\mathsf{M}_{\rho_1}[\mathbf{g}])).$$

Here, the *metric distortion loss* $\mathcal{R}: \mathbb{S}_+^L \to \mathbb{R}_+$ is defined as $\mathcal{R}(\mathsf{M}) \triangleq \mathrm{tr}(\mathsf{M}^2 - \mathsf{M} - \ln \mathsf{M})$ for $\mathsf{M} \succeq 0$. If $\gamma = 0$ and $\mathsf{M}_{\rho_0}[\mathbf{f}]$ and $\mathsf{M}_{\rho_1}[\mathbf{g}]$ are nonsingular, the objective becomes equivalent to the VAMP-2 score.

To detour the potential numerical instability in the first term of the DPNet objective, which is essentially the VAMP-2 objective, Kostic et al. [16] further proposed a relaxed objective

$$\mathcal{L}_{\mathrm{dpr}}^{(\gamma)}(\mathbf{f}, \mathbf{g}) \triangleq -\frac{\|\mathsf{T}[\mathbf{f}, \mathbf{g}]\|_{\mathrm{F}}^2}{\|\mathsf{M}_{\rho_0}[\mathbf{f}]\|_{\mathrm{op}} \|\mathsf{M}_{\rho_1}[\mathbf{g}]\|_{\mathrm{op}}} + \gamma(\mathcal{R}(\mathsf{M}_{\rho_0}[\mathbf{f}]) + \mathcal{R}(\mathsf{M}_{\rho_1}[\mathbf{g}])).$$

Kostic et al. [16] proved the following statement:

**Theorem B.2** (Consistency of DPNet objectives). *Let $\gamma \geq 0$. If $\mathcal{T}$ is compact,*

$$\mathcal{L}_{\mathrm{dpr}}^{(\gamma)}(\mathbf{f}, \mathbf{g}) \geq \mathcal{L}_{\mathrm{dp}}^{(\gamma)}(\mathbf{f}, \mathbf{g}) \geq -\sum_{i=1}^k \sigma_i(\mathcal{T})^2.$$

*The equalities are attained when $\mathbf{f}(\mathbf{x})$ and $\mathbf{g}(\mathbf{y})$ are the top-$k$ singular functions of $\mathcal{T}$. If $\mathcal{T}$ is Hilbert–Schmidt and $\sigma_L(\mathcal{T}) > \sigma_{k+1}(\mathcal{T})$ and $\gamma > 0$, the equalities are attained only if $\mathbf{f}(\mathbf{x})$ and $\mathbf{g}(\mathbf{y})$ are orthogonal rotations of top-$k$ singular functions.*

For the continuous-time dynamics with a self-adjoint Koopman generator $\mathcal{L}$, the authors proposed to use

$$\tilde{\mathcal{L}}_{\mathrm{dp}}^{(\gamma)}(\mathbf{f}) \triangleq -\mathrm{tr}(\mathsf{M}_\rho[\mathbf{f}]^\dagger \mathsf{M}_\rho[\mathbf{f}, \mathcal{L}\mathbf{f}]) + \gamma \mathcal{R}(\mathsf{M}_\rho[\mathbf{f}]).$$

# C Deferred Technical Statements and Proofs

In this section, we provide a short derivation of the low-rank approximation (LoRA) loss for completeness, and present a learning-theoretic guarantee for the LoRA objective.

## C.1 Derivation of Low-Rank Approximation Objective

**Proposition C.1.**

$$\min_{\mathbf{f}, \mathbf{g}} \left\| \mathcal{K} - \sum_{i=1}^k f_i \otimes g_i \right\|_{\mathrm{HS}}^2 = \min_{\mathbf{f}, \mathbf{g}} \mathcal{L}_{\mathrm{lora}}(\mathbf{f}, \mathbf{g}).$$

*Proof.* Recall that $\mathcal{L}_{\text{lora}}(\mathbf{f}, \mathbf{g}) \triangleq -2\operatorname{tr}(\mathsf{T}[\mathbf{f}, \mathbf{g}]) + \operatorname{tr}(\mathsf{M}_{\rho_0}[\mathbf{f}]\mathsf{M}_{\rho_1}[\mathbf{g}])$. To prove, first note that

$$\left\| \mathcal{K} - \sum_{i=1}^{k} f_i \otimes g_i \right\|_{\text{HS}}^2 - \|\mathcal{K}\|_{\text{HS}}^2 = -2\sum_{i=1}^{k}\langle f_i, \mathcal{K}g_i\rangle_{\rho_0} + \sum_{i=1}^{k}\sum_{j=1}^{k}\langle f_i, f_j\rangle_{\rho_0}\langle g_i, g_j\rangle_{\rho_1}.$$

Here, the first term can be rewritten as

$$\begin{aligned}
\sum_{i=1}^{k}\langle f_i, \mathcal{K}g_i\rangle_{\rho_0} &= \sum_{i=1}^{k}\mathbb{E}_{\rho_0(\mathbf{x})p(\mathbf{x}'\,|\,\mathbf{x})}[f_i(\mathbf{x})g_i(\mathbf{x}')] \\
&= \operatorname{tr}\big(\mathbb{E}_{\rho_0(\mathbf{x})p(\mathbf{x}'\,|\,\mathbf{x})}\big[\mathbf{f}(\mathbf{x})\mathbf{g}(\mathbf{x}')^{\mathsf{T}}\big]\big) \\
&= \operatorname{tr}(\mathsf{T}[\mathbf{f}, \mathbf{g}]).
\end{aligned}$$

Likewise, the second term is

$$\begin{aligned}
\sum_{i=1}^{k}\sum_{j=1}^{k}\langle f_i, f_j\rangle_{\rho_0}\langle g_i, g_j\rangle_{\rho_1} &= \operatorname{tr}\big(\mathbb{E}_{\rho_0(\mathbf{x})}\big[\mathbf{f}(\mathbf{x})\mathbf{f}(\mathbf{x})^{\mathsf{T}}\big]\mathbb{E}_{\rho_1(\mathbf{x}')}\big[\mathbf{g}(\mathbf{x}')\mathbf{g}(\mathbf{x}')^{\mathsf{T}}\big]\big) \\
&= \operatorname{tr}(\mathsf{M}_{\rho_0}[\mathbf{f}]\mathsf{M}_{\rho_1}[\mathbf{g}]).
\end{aligned}$$

This concludes the proof. $\qquad\square$

## C.2  Statistical-Learning-Theoretic Guarantee

We consider empirical estimates of the second moment matrices $\mathsf{T}[\mathbf{f}, \mathbf{g}]$, $\mathsf{M}_{\rho_0}[\mathbf{f}]$, $\mathsf{M}_{\rho_1}[\mathbf{g}]$.

$$\hat{\mathsf{T}}[\mathbf{f}, \mathbf{g}] \triangleq \frac{1}{N}\sum_{t=1}^{N}\mathbf{f}(\mathbf{x}_t)\mathbf{g}(\mathbf{x}_t')^{\mathsf{T}},$$

$$\hat{\mathsf{M}}_{\rho_0}[\mathbf{f}] \triangleq \frac{1}{N_0}\sum_{i=1}^{N_0}\mathbf{f}(\breve{\mathbf{x}}_i)\mathbf{f}(\breve{\mathbf{x}}_i)^{\mathsf{T}},$$

$$\hat{\mathsf{M}}_{\rho_1}[\mathbf{g}] \triangleq \frac{1}{N_1}\sum_{j=1}^{N}\mathbf{g}(\breve{\mathbf{x}}_j')\mathbf{g}(\breve{\mathbf{x}}_j')^{\mathsf{T}}.$$

Here, $(\mathbf{x}_t, \mathbf{x}_t') \sim \rho_0(\mathbf{x})p(\mathbf{x}'\,|\,\mathbf{x})$ are i.i.d. samples and $\breve{\mathbf{x}}_i \sim \rho_0(\mathbf{x})$ and $\breve{\mathbf{x}}_j' \sim \rho_1(\mathbf{x}')$ are i.i.d. samples.

Note that, if we follow the common data collection procedure, $\breve{\mathbf{x}}_i \sim \rho_0(\mathbf{x})$ and $\breve{\mathbf{x}}_j' \sim \rho_1(\mathbf{x}')$ are drawn from a single trajectory, and thus the samples $(\breve{\mathbf{x}}, \breve{\mathbf{x}}')$ cannot be independent. The independence assumption between $\{\breve{\mathbf{x}}_i\}$ and $\{\breve{\mathbf{x}}_j'\}$ are for the sake of simpler analysis. In practice, however, this can be enforced by splitting the set of given independent trajectories into two subsets and compute $\hat{\mathsf{M}}_{\rho_0}[\mathbf{f}]$ and $\hat{\mathsf{M}}_{\rho_1}[\mathbf{g}]$ with different subsets of samples.

**Theorem C.1.** *Let $\|\mathbf{f}(\mathbf{x})\| \le R$ and $\|\mathbf{g}(\mathbf{x}')\| \le R$ almost surely for some $R > 0$. Let $\mathsf{V_f} \triangleq \mathbb{E}_{\rho_0(\mathbf{x})}[(\mathbf{f}(\mathbf{x})\mathbf{f}(\mathbf{x})^{\mathsf{T}} - \mathsf{M}_{\rho_0}[\mathbf{f}])^2] \in \mathbb{R}^{k \times k}$ and $\mathsf{V_g} \triangleq \mathbb{E}_{\rho_1(\mathbf{x}')}[(\mathbf{g}(\mathbf{x}')\mathbf{g}(\mathbf{x}')^{\mathsf{T}} - \mathsf{M}_{\rho_1}[\mathbf{g}])^2] \in \mathbb{R}^{k \times k}$. Then, with probability at least $1 - \delta$, we have*

$$\begin{aligned}
|\hat{\mathcal{L}} - \mathcal{L}| \le R^2\Bigg\{ &\sqrt{\frac{16}{N}\log\frac{6}{\delta}} + \frac{8}{3N}\log\frac{6}{\delta} \\
&+ \sqrt{\frac{2\|\mathsf{V_f}\|_{\text{op}}}{N_0}\log\frac{12r(\mathsf{V_f})}{\delta}} + \frac{2R^2}{3N_0}\log\frac{12r(\mathsf{V_f})}{\delta} \\
&+ \sqrt{\frac{2\|\mathsf{V_g}\|_{\text{op}}}{N_1}\log\frac{12r(\mathsf{V_g})}{\delta}} + \frac{2R^2}{3N_1}\log\frac{12r(\mathsf{V_g})}{\delta}\Bigg\}.
\end{aligned}$$

To prove this, we will invoke Bernstein inequalities for bounded random variables and "bounded" self-adjoint matrices:

**Lemma C.1** (Bernstein's inequality [38, Theorem 1.6.1]). Let $X_1, \ldots, X_n$ be i.i.d. copies of a random variable $X$ such that $|X| \leq u$ almost surely, $\mathbb{E}[X] = 0$, and $\mathbb{E}[X^2] \leq \sigma^2$. Then, for any $\delta \in (0, 1)$, with probability at least $1 - \delta$,

$$\left| \frac{1}{n} \sum_{i=1}^{n} X_i \right| \leq \frac{4u}{3n} \log \frac{2}{\delta} + \sqrt{\frac{4\sigma^2}{n} \log \frac{2}{\delta}}.$$

**Lemma C.2** (Matrix Bernstein inequality with intrinsic dimension [26], [38, Theorem 7.7.1]). Let $\mathsf{A}_1, \ldots, \mathsf{A}_n$ be i.i.d. copies of a random self-adjoint matrix $\mathsf{A}$, which satisfies $\|\mathsf{A}\|_{\mathrm{op}} \leq c$ almost surely, $\mathbb{E}[\mathsf{A}] = 0$, and $\mathbb{E}[\mathsf{A}^2] = \mathsf{V} \succeq 0$. Let $r(\mathsf{V}) \triangleq \frac{\mathrm{tr}(\mathsf{V})}{\|\mathsf{V}\|_{\mathrm{op}}}$ denote the *intrinsic dimension* of $\mathsf{V}$. Then, for any $\delta \in (0, 1)$, with probability at least $1 - \delta$,

$$\left\| \frac{1}{n} \sum_{i=1}^{n} \mathsf{A}_i \right\|_{\mathrm{op}} \leq \frac{2c}{3n} \log \frac{4r(\mathsf{V})}{\delta} + \sqrt{\frac{2\|\mathsf{V}\|_{\mathrm{op}}}{n} \log \frac{4r(\mathsf{V})}{\delta}}.$$

We are now ready to prove Theorem C.1.

*Proof of Theorem C.1.* Consider

$$|\hat{\mathcal{L}} - \mathcal{L}| = \left| \mathrm{tr}(-2\hat{\mathsf{T}}[\mathbf{f}, \mathbf{g}] + \hat{\mathsf{M}}_{\rho_0}[\mathbf{f}]\hat{\mathsf{M}}_{\rho_1}[\mathbf{g}]) - \mathrm{tr}(-2\mathsf{T}[\mathbf{f}, \mathbf{g}] + \mathsf{M}_{\rho_0}[\mathbf{f}]\mathsf{M}_{\rho_1}[\mathbf{g}]) \right|$$

$$\leq 2|\mathrm{tr}(\hat{\mathsf{T}}[\mathbf{f}, \mathbf{g}] - \mathsf{T}[\mathbf{f}, \mathbf{g}])| + |\mathrm{tr}(\hat{\mathsf{M}}_{\rho_0}[\mathbf{f}]\hat{\mathsf{M}}_{\rho_1}[\mathbf{g}] - \mathsf{M}_{\rho_0}[\mathbf{f}]\mathsf{M}_{\rho_1}[\mathbf{g}])|.$$

For the first term $|\mathrm{tr}(\hat{\mathsf{T}}[\mathbf{f}, \mathbf{g}] - \mathsf{T}[\mathbf{f}, \mathbf{g}])|$, note that

$$\mathrm{tr}(\hat{\mathsf{T}}[\mathbf{f}, \mathbf{g}] - \mathsf{T}[\mathbf{f}, \mathbf{g}]) = \frac{1}{N} \sum_{t=1}^{N} \mathbf{f}(\mathbf{x}_t)^\intercal \mathbf{g}(\mathbf{x}'_t) - \mathbb{E}_{\rho_0(\mathbf{x})p(\mathbf{x}'|\mathbf{x})}\left[ \mathbf{f}(\mathbf{x})^\intercal \mathbf{g}(\mathbf{x}') \right].$$

Here, we note that

$$|\mathbf{f}(\mathbf{x})^\intercal \mathbf{g}(\mathbf{x}')| \leq \|\mathbf{f}(\mathbf{x})\| \cdot \|\mathbf{g}(\mathbf{x}')\| \leq R^2,$$
$$(\mathbf{f}(\mathbf{x})^\intercal \mathbf{g}(\mathbf{x}'))^2 \leq R^4.$$

Hence, we can apply Bernstein's inequality in Lemma C.1 for $(\mathbf{f}(\mathbf{x}_t)^\intercal \mathbf{g}(\mathbf{x}'_t))_{t=1}^{N}$ with $u \leftarrow R^2$, $\sigma^2 \leftarrow R^4$, and $\delta \leftarrow \delta'$, where $\delta'$ is to be decided at the end of proof.

Now, we wish to bound $|\mathrm{tr}(\hat{\mathsf{M}}_{\rho_0}[\mathbf{f}]\hat{\mathsf{M}}_{\rho_1}[\mathbf{g}] - \mathsf{M}_{\rho_0}[\mathbf{f}]\mathsf{M}_{\rho_1}[\mathbf{g}])|$.

Let $\Delta_{\mathbf{f}} \triangleq \hat{\mathsf{M}}_{\rho_0}[\mathbf{f}] - \mathsf{M}_{\rho_0}[\mathbf{f}]$ and $\Delta_{\mathbf{g}} \triangleq \hat{\mathsf{M}}_{\rho_1}[\mathbf{g}] - \mathsf{M}_{\rho_1}[\mathbf{g}]$. Then,

$$|\mathrm{tr}(\hat{\mathsf{M}}_{\rho_0}[\mathbf{f}]\hat{\mathsf{M}}_{\rho_1}[\mathbf{g}] - \mathsf{M}_{\rho_0}[\mathbf{f}]\mathsf{M}_{\rho_1}[\mathbf{g}])| = |\mathrm{tr}(\Delta_{\mathbf{f}}\mathsf{M}_{\rho_1}[\mathbf{g}] + \hat{\mathsf{M}}_{\rho_0}[\mathbf{f}]\Delta_{\mathbf{g}})|$$

$$\leq |\mathrm{tr}(\Delta_{\mathbf{f}}\mathsf{M}_{\rho_1}[\mathbf{g}])| + |\mathrm{tr}(\hat{\mathsf{M}}_{\rho_0}[\mathbf{f}]\Delta_{\mathbf{g}})|$$

$$\overset{(a)}{\leq} \|\Delta_{\mathbf{f}}\|_{\mathrm{op}} \mathrm{tr}(\mathsf{M}_{\rho_1}[\mathbf{g}]) + \|\Delta_{\mathbf{g}}\|_{\mathrm{op}} \mathrm{tr}(\hat{\mathsf{M}}_{\rho_0}[\mathbf{f}])$$

$$= \|\Delta_{\mathbf{f}}\|_{\mathrm{op}} \mathbb{E}_{\pi}[\|\mathbf{g}(\mathbf{x}')\|^2] + \|\Delta_{\mathbf{g}}\|_{\mathrm{op}} \frac{1}{N_1} \sum_{j=1}^{N_1} \|\mathbf{g}(\mathbf{x}'_j)\|^2$$

$$\overset{(b)}{\leq} R^2 \left( \|\Delta_{\mathbf{f}}\|_{\mathrm{op}} + \|\Delta_{\mathbf{g}}\|_{\mathrm{op}} \right).$$

where $(a)$ follows from the inequality $|\mathrm{tr}(\mathsf{AB})| \leq \|\mathsf{A}\|_{\mathrm{op}} \mathrm{tr}(\mathsf{B})$ for square matrices $\mathsf{A}, \mathsf{B}$ such that $\mathsf{B} \succeq 0$, and $(b)$ from the boundedness assumption, i.e., $\|\mathbf{f}(\mathbf{x})\| \leq R$ and $\|\mathbf{g}(\mathbf{x}')\| \leq R$ almost surely. Here, we can apply the matrix Bernstein inequality in Lemma C.2 to bound $\|\Delta_{\mathbf{f}}\|_{\mathrm{op}}$ with $\mathsf{A} \leftarrow \mathbf{f}(\mathbf{x})\mathbf{f}(\mathbf{x})^\intercal - \mathsf{M}_{\rho_0}[\mathbf{f}]$ with $\delta \leftarrow 1 - \delta'$, and similarly for $\|\Delta_{\mathbf{g}}\|_{\mathrm{op}}$.

Finally, by applying the union bound on the Bernstein inequality and two matrix Bernstein inequalities with $\delta' = \delta/3$, we can conclude the desired bound. $\qquad\square$

# D    Special Case of Finite-Rank Koopman Operator

If a given operator is of finite rank, we can analyze the operator using finite-dimensional matrices of the same dimension. We can develop a computational procedure for SVD and eigen-analysis under the finite-rank assumption, and apply the tools once we approximate the target operator using a low rank expansion. Some of the results established in this section are used later when (numerically) computing the ground truth characteristics of the noisy logistic map studied in Appendix G.1.

Suppose that the Koopman operator $\mathcal{K}\colon L^2_{\rho_1}(\mathcal{X}) \to L^2_{\rho_0}(\mathcal{X})$ of our interest is of finite rank. Specifically, the corresponding kernel can be expressed in the following factorized form:

$$k(\mathbf{x}, \mathbf{x}') = \frac{p(\mathbf{x}' \mid \mathbf{x})}{\rho_1(\mathbf{x}')} = \boldsymbol{\alpha}(\mathbf{x})^\mathsf{T}\boldsymbol{\beta}(\mathbf{x}') = \sum_{i=1}^r \alpha_i(\mathbf{x})\beta_i(\mathbf{x}').$$

Due to the separable form, the kernel has a rank at most $r$. We aim to find its SVD as follows:

$$k(\mathbf{x}, \mathbf{x}') = \sum_{i=1}^r \sigma_i \phi_i(\mathbf{x})\psi_i(\mathbf{x}'), \tag{13}$$

where the functions satisfy the orthogonality conditions: $\langle \phi_i, \phi_j \rangle_{\rho_0} = \langle \psi_i, \psi_j \rangle_{\rho_1} = \delta_{ij}$. The singular values $\sigma_1 = 1 \geq \sigma_2 \geq \ldots \geq \sigma_r \geq 0$. The first singular functions $\phi_1$ and $\psi_1$, corresponding to the trivial mode $\sigma_1 = 1$, are constant functions.

## D.1    Singular Value Decomposition

We can compute the singular value decomposition (SVD) using $\mathsf{M}_{\rho_0}[\boldsymbol{\alpha}]$ and $\mathsf{M}_{\rho_1}[\boldsymbol{\beta}]$ as follows.

**Theorem D.1.** *The SVD of the matrix*

$$\mathsf{S}^{\mathsf{sqrt}}_{\rho_0, \rho_1}[\boldsymbol{\alpha}, \boldsymbol{\beta}] \triangleq (\mathsf{M}_{\rho_0}[\boldsymbol{\alpha}])^{1/2}(\mathsf{M}_{\rho_1}[\boldsymbol{\beta}])^{1/2}$$

*shares the same spectrum, i.e., there exist orthonormal matrices* $\mathsf{U} \in \mathbb{R}^{k \times r}$ *and* $\mathsf{V} \in \mathbb{R}^{k \times r}$ *and* $\Sigma = \mathsf{diag}(\sigma_1, \ldots, \sigma_r)$ *such that*

$$\mathsf{S}^{\mathsf{sqrt}}_{\rho_0, \rho_1}[\boldsymbol{\alpha}, \boldsymbol{\beta}] = \mathsf{U}\Sigma\mathsf{V}^\mathsf{T} = \sum_{i=1}^r \sigma_i \mathbf{u}_i \mathbf{v}_i^\mathsf{T}.$$

*The ordered, normalized singular functions of the kernel* $k(\mathbf{x}, \mathbf{x}')$ *are given as*

$$\boldsymbol{\phi}(\mathbf{x}) = \mathsf{U}^\mathsf{T}(\mathsf{M}_{\rho_0}[\boldsymbol{\alpha}])^{-1/2}\boldsymbol{\alpha}(\mathbf{x}), \tag{14}$$

$$\boldsymbol{\psi}(\mathbf{x}') = \mathsf{V}^\mathsf{T}(\mathsf{M}_{\rho_1}[\boldsymbol{\beta}])^{-1/2}\boldsymbol{\beta}(\mathbf{x}'). \tag{15}$$

*Proof.* First of all, it is easy to check that $\mathsf{M}_{\rho_0}[\boldsymbol{\phi}] = \mathsf{M}_{\rho_1}[\boldsymbol{\psi}] = \mathsf{I}$. Next, we can show Eq. (13). To show this,

$$\sum_{i=1}^r \sigma_i \phi_i(\mathbf{x})\psi_i(\mathbf{x}') = \boldsymbol{\phi}(\mathbf{x})^\mathsf{T}\Sigma\boldsymbol{\psi}(\mathbf{x}')$$

$$= (\mathsf{U}^\mathsf{T}(\mathsf{M}_{\rho_0}[\boldsymbol{\alpha}])^{-1/2}\boldsymbol{\alpha}(\mathbf{x}))^\mathsf{T}\mathsf{U}^\mathsf{T}\mathsf{S}^{\mathsf{sqrt}}_{\rho_0, \rho_1}[\boldsymbol{\alpha}, \boldsymbol{\beta}]\mathsf{V}(\mathsf{V}^\mathsf{T}(\mathsf{M}_{\rho_1}[\boldsymbol{\beta}])^{-1/2}\boldsymbol{\beta}(\mathbf{x}'))$$

$$= \boldsymbol{\alpha}(\mathbf{x})^\mathsf{T}(\mathsf{M}_{\rho_0}[\boldsymbol{\alpha}])^{-1/2}((\mathsf{M}_{\rho_0}[\boldsymbol{\alpha}])^{1/2}(\mathsf{M}_{\rho_1}[\boldsymbol{\beta}])^{1/2})(\mathsf{M}_{\rho_1}[\boldsymbol{\beta}])^{-1/2}\boldsymbol{\beta}(\mathbf{x}')$$

$$= \boldsymbol{\alpha}(\mathbf{x})^\mathsf{T}\boldsymbol{\beta}(\mathbf{x}')$$

$$= k(\mathbf{x}, \mathbf{x}').$$

To show $(\mathcal{K}\phi_i)(\mathbf{x}') = \sigma_i \psi_i(\mathbf{x}')$, consider

$$(\mathcal{K}\phi_i)(\mathbf{x}') \triangleq \int k(\mathbf{x}, \mathbf{x}')\phi_i(\mathbf{x})\rho_0(\mathbf{x})dx$$

$$= \boldsymbol{\beta}(\mathbf{x}')^\mathsf{T} \int \boldsymbol{\alpha}(\mathbf{x})(\mathbf{u}_i^\mathsf{T}(\mathsf{M}_{\rho_0}[\boldsymbol{\alpha}])^{-1/2}\boldsymbol{\alpha}(\mathbf{x}))\rho_0(\mathbf{x})dx$$

$$= \boldsymbol{\beta}(\mathbf{x}')^\mathsf{T} \int \boldsymbol{\alpha}(\mathbf{x})\boldsymbol{\alpha}(\mathbf{x})^\mathsf{T} \rho_0(\mathbf{x})dx (\mathsf{M}_{\rho_0}[\boldsymbol{\alpha}])^{-1/2}\mathbf{u}_i$$

$$= \boldsymbol{\beta}(\mathbf{x}')^\mathsf{T} (\mathsf{M}_{\rho_0}[\boldsymbol{\alpha}])^{1/2}\mathbf{u}_i$$

$$= \boldsymbol{\beta}(\mathbf{x}')^\mathsf{T} (\mathsf{M}_{\rho_1}[\boldsymbol{\beta}])^{-1/2}(\mathsf{S}^{\mathsf{sqrt}}_{\rho_0,\rho_1}[\boldsymbol{\alpha},\boldsymbol{\beta}])^\mathsf{T}\mathbf{u}_i$$

$$= \sigma_i \boldsymbol{\beta}(\mathbf{x}')^\mathsf{T} (\mathsf{M}_{\rho_1}[\boldsymbol{\beta}])^{-1/2}\mathbf{v}_i$$

$$= \sigma_i \psi_i(\mathbf{x}').$$

We can show $(\mathcal{K}^*\psi_i)(\mathbf{x}) = \sigma_i\phi_i(\mathbf{x})$ by the same reasoning. □

## D.2 Eigen-analysis

We can compute the eigenvalues and eigenfunctions of the kernel by eigen-analyzing the matrices $\mathsf{M}_{\rho_1}[\boldsymbol{\beta},\boldsymbol{\alpha}]$ and $\mathsf{M}_{\rho_0}[\boldsymbol{\beta},\boldsymbol{\alpha}]$.

**Theorem D.2.** *Let* $\mathbf{w} \in \mathbb{C}^k$ *be a right eigenvector of* $\mathsf{M}_{\rho_1}[\boldsymbol{\beta},\boldsymbol{\alpha}]$ *with eigenvalue* $\lambda \in \mathbb{C}$, *i.e.,* $\mathsf{M}_{\rho_1}[\boldsymbol{\beta},\boldsymbol{\alpha}]\mathbf{w} = \lambda\mathbf{w}$. *If we define* $\eta(\mathbf{x}') \triangleq \mathbf{w}^\mathsf{T}\boldsymbol{\alpha}(\mathbf{x}')$, *then* $\eta$ *is a right eigenfunction of* $\mathcal{K}$ *with eigenvalue* $\lambda$, *i.e.,* $\mathcal{K}\eta = \lambda\eta$.

*Proof.* Since $k(\mathbf{x},\mathbf{x}') = \boldsymbol{\alpha}(\mathbf{x})^\mathsf{T}\boldsymbol{\beta}(\mathbf{x}')$, we have

$$(\mathcal{K}\eta)(\mathbf{x}) = \int k(\mathbf{x},\mathbf{x}')\eta(\mathbf{x}')\rho_1(\mathbf{x}')d\mathbf{x}'$$

$$= \boldsymbol{\alpha}(\mathbf{x})^\mathsf{T} \int \boldsymbol{\beta}(\mathbf{x}')\boldsymbol{\alpha}(\mathbf{x}')^\mathsf{T}\mathbf{w}\rho_1(\mathbf{x}')d\mathbf{x}'$$

$$= \boldsymbol{\alpha}(\mathbf{x})^\mathsf{T}\mathsf{M}_{\rho_1}[\boldsymbol{\beta},\boldsymbol{\alpha}]\mathbf{w}$$

$$= \lambda\boldsymbol{\alpha}(\mathbf{x})^\mathsf{T}\mathbf{w}$$

$$= \lambda\eta(\mathbf{x}). \qquad \square$$

**Theorem D.3.** *Let* $\mathbf{z} \in \mathbb{C}^k$ *be a left eigenvector of* $\mathsf{M}_{\rho_0}[\boldsymbol{\beta},\boldsymbol{\alpha}]$ *with eigenvalue* $\lambda \in \mathbb{C}$, *i.e.,* $\mathbf{z}^*\mathsf{M}_{\rho_0}[\boldsymbol{\beta},\boldsymbol{\alpha}] = \lambda\mathbf{z}^*$. *If we define* $\zeta(\mathbf{x}') \triangleq \mathbf{z}^\mathsf{T}\boldsymbol{\beta}(\mathbf{x}')$, *then* $\zeta$ *is a left eigenfunction of* $\mathcal{K}$ *with eigenvalue* $\lambda$, *i.e.,* $\mathcal{K}^*\zeta = \bar{\lambda}\zeta$.

*Proof.* We have

$$(\mathcal{K}^*\zeta)(\mathbf{x}) = \int k(\mathbf{x},\mathbf{x}')\zeta(\mathbf{x})\rho_0(\mathbf{x})d\mathbf{x}$$

$$= \boldsymbol{\beta}(\mathbf{x}')^\mathsf{T} \int \boldsymbol{\alpha}(\mathbf{x})\boldsymbol{\beta}(\mathbf{x})^\mathsf{T}\mathbf{z}\rho_0(\mathbf{x})d\mathbf{x}$$

$$= \boldsymbol{\beta}(\mathbf{x}')^\mathsf{T}(\mathsf{M}_{\rho_0}[\boldsymbol{\beta},\boldsymbol{\alpha}])^\mathsf{T}\mathbf{z}$$

$$= \bar{\lambda}\boldsymbol{\beta}(\mathbf{x}')^\mathsf{T}\mathbf{z}$$

$$= \bar{\lambda}\zeta(\mathbf{x}'). \qquad \square$$

Interestingly, this implies that spectrum of $\mathsf{M}_{\rho_1}[\boldsymbol{\beta},\boldsymbol{\alpha}]$ and $\mathsf{M}_{\rho_0}[\boldsymbol{\beta},\boldsymbol{\alpha}]$ must be identical.

Moreover, if $\mathbf{y}_o$ denotes the Perron–Frobenius eigenvector of $\mathsf{M}_{\rho_0}[\boldsymbol{\beta},\boldsymbol{\alpha}]$ (i.e., the left eigenvector with eigenvalue 1), then $g_o(\mathbf{x}) \triangleq \mathbf{y}_o^\mathsf{T}\boldsymbol{\beta}(\mathbf{x})$ is the stationary distribution.

# E Extended DMD and Multi-Step Prediction

In this section, we illustrate the extended DMD (EDMD) procedure [46] using basic ideas as elementary as linear regression. Suppose that we are given top-$k$ left and right singular functions $\mathbf{f}(\cdot)$ (encoder) and $\mathbf{g}(\cdot)$ (lagged encoder), respectively. If $h \in \mathsf{span}(\mathbf{f}) \triangleq \mathsf{span}\{f_1,\ldots,f_k\}$, then there exists some $\mathbf{z} \in \mathbb{R}^k$ such that $h(\mathbf{x}) = \mathbf{z}^\mathsf{T}\mathbf{f}(\mathbf{x})$. Then, we denote by $\mathcal{K}_\mathbf{f}$ the restriction of the Koopman operator onto the span, which acts on $h$ as

$$(\mathcal{K}_\mathbf{f}h)(\mathbf{x}) = \mathbf{f}(\mathbf{x})^\mathsf{T}\mathsf{K}_\mathbf{f}\mathbf{z}$$

for some $\mathsf{K}_\mathbf{f} \in \mathbb{R}^{k \times k}$. With finite data, there are two issues in this picture: (1) How can we compute $\mathsf{K}_\mathbf{f}$ from data? (2) Given $h \in \mathsf{span}(\mathbf{f})$, how can we compute the corresponding $\mathbf{z} \in \mathbb{R}^k$?

- For the first question, the EDMD aims to find the best finite-dimensional approximation of the Koopman operator $\mathcal{K}$ restricted on $\mathsf{span}(\mathbf{f})$, in the sense that $(\mathcal{K}\mathbf{f})(\mathbf{x}) = \mathbb{E}_{p(\mathbf{x}' \mid \mathbf{x})}[\mathbf{f}(\mathbf{x}')] \approx \mathsf{K}\mathbf{f}(\mathbf{x})$. As a natural choice, we can choose the ordinary least square solution that solves $\mathbf{f}(\mathbf{x}') = \mathsf{K}\mathbf{f}(\mathbf{x})$ for $(\mathbf{x}, \mathbf{x}') \sim \rho_0(\mathbf{x})p(\mathbf{x}' \mid \mathbf{x})$, where

$$\hat{\mathsf{K}}_\mathbf{f}^{\mathsf{ols}} \triangleq \mathsf{F}^+ \mathsf{F}' = (\mathsf{F}^\mathsf{T}\mathsf{F})^{-1}\mathsf{F}^\mathsf{T}\mathsf{F}' = (\hat{\mathsf{M}}_{\rho_0}[\mathbf{f}])^{-1}\hat{\mathsf{T}}[\mathbf{f}],$$

where we denote the data matrices by

$$\mathsf{F} \triangleq [\mathbf{f}(\mathbf{x}_1) \ldots \mathbf{f}(\mathbf{x}_N)]^\mathsf{T} \in \mathbb{R}^{N \times k} \quad \text{and} \quad \mathsf{F}' \triangleq [\mathbf{f}(\mathbf{x}_1') \ldots \mathbf{f}(\mathbf{x}_N')]^\mathsf{T} \in \mathbb{R}^{N \times k}.$$

Note that $\hat{\mathsf{M}}_{\rho_0}[\mathbf{f}]$ and $\hat{\mathsf{T}}[\mathbf{f}]$ are the empirical estimates of the second moment matrices $\mathsf{M}_{\rho_0}[\mathbf{f}]$ and $\mathsf{T}[\mathbf{f}]$.

- For the second point, we can again view this as a linear regression problem, since

$$h(\mathbf{x}_i) = \mathbf{z}^\mathsf{T}\mathbf{f}(\mathbf{x}_i) \quad \text{for } i = 1, \ldots, N.$$

Hence, we can define the best $\mathbf{z}$ again as the OLS solution $\mathbf{z}_h^{\mathsf{ols}}[\mathbf{f}] \triangleq \mathsf{F}^+\mathsf{H} = (\hat{\mathsf{M}}_{\rho_0}[\mathbf{f}])^{-1}\mathsf{F}^\mathsf{T}\mathsf{H}$, where

$$\mathsf{H} \triangleq [h(\mathbf{x}_1), \ldots, h(\mathbf{x}_N)]^\mathsf{T} \in \mathbb{R}^N.$$

Now, suppose that we already computed $\hat{\mathsf{T}}[\mathbf{f}, \mathbf{g}]$ and $\hat{\mathbf{z}}_h$ from data. Then, we can approximate the multi-step prediction as

$$\mathbb{E}_{p(\mathbf{x}_t \mid \mathbf{x}_0)}[h(\mathbf{x}_t)] \approx \mathbf{f}(\mathbf{x}_0)^\mathsf{T}\hat{\mathsf{T}}[\mathbf{f}, \mathbf{g}]^t\hat{\mathbf{z}}_h. \tag{16}$$

The same logic applies to the subspace spanned by the right singular functions $\psi(\cdot)$.

# F    On Implementation

In this section, we explain how we can implement nesting with automatic differentiation and provide a pseudocode in PyTorch [30].

## F.1    Implementation of Nesting Techniques

We can implement the sequential nesting by computing the derivative of the following objective using automatic differentiation:

$$\mathcal{L}_{\mathsf{lora}}^{\mathsf{seq}}(\mathbf{f}, \mathbf{g}) \triangleq -2\,\mathrm{tr}(\mathsf{M}_{\rho_0}^{\mathsf{seq}}[\mathbf{f}, \mathcal{K}\mathbf{g}]) + \mathrm{tr}(\mathsf{M}_{\rho_0}^{\mathsf{seq}}[\mathbf{f}]\mathsf{M}_{\rho_1}^{\mathsf{seq}}[\mathbf{g}]),$$

where we define a partially stop-gradient second moment matrix

$$\mathsf{M}_\rho^{\mathsf{seq}}[\mathbf{f}, \mathbf{g}] \triangleq \begin{bmatrix} \langle f_1, f_1 \rangle_\rho & \langle \mathsf{sg}[f_1], f_2 \rangle_\rho & \langle \mathsf{sg}[f_1], f_3 \rangle_\rho & \cdots & \langle \mathsf{sg}[f_1], f_k \rangle_\rho \\ \langle f_2, \mathsf{sg}[f_1] \rangle_\rho & \langle f_2, f_2 \rangle_\rho & \langle \mathsf{sg}[f_2], f_3 \rangle_\rho & \cdots & \langle \mathsf{sg}[f_2], f_k \rangle_\rho \\ \langle f_3, \mathsf{sg}[f_1] \rangle_\rho & \langle f_3, \mathsf{sg}[f_2] \rangle_\rho & \langle f_3, f_3 \rangle_\rho & \cdots & \langle \mathsf{sg}[f_3], f_k \rangle_\rho \\ \vdots & \vdots & & \ddots & \vdots \\ \langle f_k, \mathsf{sg}[f_1] \rangle_\rho & \langle f_k, \mathsf{sg}[f_2] \rangle_\rho & \langle f_k, \mathsf{sg}[f_3] \rangle_\rho & \cdots & \langle f_k, f_k \rangle_\rho \end{bmatrix}.$$

This can be implemented efficiently with almost no additional computation. We note that the implementation of the sequentially nested LoRA we provide here is a more direct and simpler version than the implementation of NeuralSVD in [32], which implemented the sequential nesting by a custom gradient.

The joint nesting can also be implemented in an efficient manner via the matrix mask, as explained in [32]. Define the *matrix mask* $\mathsf{P} \in \mathbb{R}^{k \times k}$ as $\mathsf{P}_{ij} = \mathsf{m}_{\max\{i,j\}}$ with $\mathsf{m}_i \triangleq \sum_{j=i}^k \alpha_j$. Then, $\mathcal{L}_{\mathsf{lora}}^{\mathsf{joint}}(\mathbf{f}, \mathbf{g}; \boldsymbol{\alpha}) \triangleq \sum_{i=1}^k \alpha_i \mathcal{L}_{\mathsf{lora}}(\mathbf{f}_{1:i}, \mathbf{g}_{1:i})$ Then, we can write

$$\mathcal{L}_{\mathsf{lora}}^{\mathsf{joint}}(\mathbf{f}, \mathbf{g}; \boldsymbol{\alpha}) = \sum_{i=1}^k \alpha_i \mathcal{L}_{\mathsf{lora}}(\mathbf{f}_{1:i}, \mathbf{g}_{1:i}) = \mathrm{tr}\Big(\mathsf{P} \odot \Big(-2\mathsf{M}_{\rho_0}[\mathbf{f}, \mathcal{K}\mathbf{g}] + \mathsf{M}_{\rho_0}[\mathbf{f}]\mathsf{M}_{\rho_1}[\mathbf{g}]\Big)\Big).$$

## F.2 Pseudocode for LoRA Loss with Nesting

Based on the nesting techniques described above, here we provide a simple and efficient PyTorch [30] implementation of the NestedLoRA (i.e., LoRA with nesting) objective.

```python
class NestedLoRALoss:
    def __init__(
        self,
        use_learned_svals=False,
        nesting=None,
        n_modes=None,
    ):
        self.use_learned_svals = use_learned_svals
        assert nesting in [None, 'seq', 'jnt']
        self.nesting = nesting
        if self.nesting == 'jnt':
            assert n_modes is not None
            self.vec_mask, self.mat_mask = get_joint_nesting_masks(
                weights=np.ones(n_modes) / n_modes,
            )
        else:
            self.vec_mask, self.mat_mask = None, None
        self.kostic_regularization = kostic_regularization

    def __call__(self, f: torch.Tensor, g: torch.Tensor):
        # f: [b, k]
        # g: [b, k]

        if self.nesting == 'jnt':
            # \sum_{i=1}^k <f_i, T g_i >
            corr_term = -2 * (self.vec_mask.to(f.device) * f * g).mean(0).sum()
            # \sum_{i=1}^k \sum_{j=1}^k <f_i, f_j> <g_i, g_j>
            # = tr ( cov (f_{1:k}) * cov(g_{1:k}) )
            M_f = compute_second_moment(f)
            M_g = compute_second_moment(g)
            metric_term = (self.mat_mask.to(f.device) * M_f * M_g).sum()

        else:
            # \sum_{i=1}^k <f_i, T g_i >
            corr_term = -2 * (f * g).mean(0).sum()
            # \sum_{i=1}^k \sum_{j=1}^k <f_i, f_j> <g_i, g_j>
            # = tr ( cov (f_{1:k}) * cov(g_{1:k}) )
            M_f = compute_second_moment(f, seq_nesting=self.nesting == 'seq')
            M_g = compute_second_moment(g, seq_nesting=self.nesting == 'seq')
            metric_term = (M_f * M_g).sum()

        return corr_term + metric_term

def compute_second_moment(
        f: torch.Tensor,
        g: torch.Tensor = None,
        seq_nesting: bool = False
    ) -> torch.Tensor:
    """
    compute (optionally sequentially nested) second-moment matrix
        M_ij = <f_i, g_j>
    with partial stop-gradient handling when seq_nesting is True.

    args
    ----
    f : (n, k) tensor
    g : (n, k) tensor or None
    seq_nesting : bool
    """
    if g is None:
```

```
61        g = f
62    n = f.shape[0]
63    if not seq_nesting:
64        return (f.T @ g) / n
65    else:
66        # partial stop gradient
67        # lower-triangular: <f_i, sg[g_j]> for i > j
68        lower = torch.tril(f.T @ g.detach(), diagonal=-1)
69        # upper-triangular: <sg[f_i], g_j> for i < j
70        upper = torch.triu(f.detach().T @ g, diagonal=+1)
71        # diagonal:         <f_i, g_i>    (no stop-grad)
72        diag  = torch.diag((f * g).sum(dim=0))
73        return (lower + diag + upper) / n
74
75 def get_joint_nesting_masks(weights: np.ndarray):
76    vector_mask = list(np.cumsum(list(weights)[::-1])[::-1])
77    vector_mask = torch.tensor(np.array(vector_mask)).float()
78    matrix_mask = torch.minimum(
79        vector_mask.unsqueeze(1), vector_mask.unsqueeze(1).T
80    ).float()
81    return vector_mask, matrix_mask
```

# G   Deferred Details on Experiments

In this section, we provide details for each experiment. See Table 3 for an overview of the benchmark problems from [16]. Our implementation builds upon the codebase of [16][6], the kooplearn package[7], and the codebase of [32][8].

Table 3: Overview of the experiments.

| Examples | Time | Spectral complexity | Stationarity |
|----------|------|---------------------|--------------|
| Noisy logistic map | discrete | non-normal | (nearly) stationary |
| Ordered MNIST | discrete | normal | stationary |
| 1D SDE | continuous | self-adjoint | (nearly) stationary |
| Molecular dynamics | continuous (discretized) | normal | non-stationary |

## G.1   Noisy Logistic Map

We consider a noisy logistic map defined as

$$X_{t+1} = (rX_t(1 - X_t) + \xi_t) \bmod 1$$

for $\xi_t \sim p(\xi)$, where $p(\xi) \triangleq C_N \cos^N(\pi\xi)$ is the order $N$ trigonometric noise over $\xi \in [-0.5, 0.5]$ [29], where $C_N \triangleq \pi/B(\frac{N+1}{2}, \frac{1}{2})$ is the (reciprocal) normalization constant.

Although the dynamics is non-normal, the structure of the polynomial trigonometric noise ensures that the associated kernel is of finite rank $N + 1$. Concretely, the transition density can be written as

$$p(x'|x) = \sum_{i=0}^{N} \alpha_i(x)\breve{\beta}_i(x'),$$

where

$$\breve{\beta}_i(x) \triangleq \sqrt{C_N \binom{N}{i}} \cos^i(\pi x) \sin^{N-i}(\pi x),$$

---

[6]https://github.com/pietronvll/DPNets
[7]https://kooplearn.readthedocs.io/
[8]https://github.com/jongharyu/neural-svd

$$\alpha_i(x) \triangleq \breve{\beta}_i(F(x)).$$

Let $\beta_i(x') \triangleq \frac{\breve{\beta}_i(x')}{\rho_1(x')}$ such that we can write $\frac{p(x'|x)}{\rho_1(x')} = \boldsymbol{\alpha}(x)^\mathsf{T}\boldsymbol{\beta}(x').$[9]

**Computation of Ground-Truth Properties.** Since the operator is of finite rank, the underlying singular functions and eigenfunctions can be computed numerically, using the theory developed in Appendix D. Let

$$\mathsf{M}_\pi[\boldsymbol{\phi}, \boldsymbol{\alpha}] \triangleq \mathbb{E}_\pi[\boldsymbol{\phi}(\mathbf{x})\boldsymbol{\alpha}(\mathbf{x})^\mathsf{T}],$$
$$\mathsf{M}_\pi[\boldsymbol{\phi}, \boldsymbol{\beta}] \triangleq \mathbb{E}_\pi[\boldsymbol{\phi}(\mathbf{x})\boldsymbol{\beta}(\mathbf{x})^\mathsf{T}].$$

Note that

$$\mathsf{M}_\pi[\boldsymbol{\phi}, \boldsymbol{\alpha}]\mathsf{M}_\pi[\boldsymbol{\phi}, \boldsymbol{\beta}]^\mathsf{T} = \mathbb{E}_{\rho_0(\mathbf{x})\rho_1(\mathbf{x}')}\Big[\boldsymbol{\phi}(\mathbf{x})\boldsymbol{\alpha}(\mathbf{x})^\mathsf{T}\boldsymbol{\beta}(\mathbf{x}')\boldsymbol{\phi}(\mathbf{x}')^\mathsf{T}\Big]$$
$$= \mathbb{E}_{\rho_0(\mathbf{x})\rho_1(\mathbf{x}')}\Big[\boldsymbol{\phi}(\mathbf{x})k(\mathbf{x}, \mathbf{x}')\boldsymbol{\phi}(\mathbf{x}')^\mathsf{T}\Big]$$
$$= \mathbb{E}_{\rho_0(\mathbf{x})p(\mathbf{x}'\,|\,\mathbf{x})}[\boldsymbol{\phi}(\mathbf{x})\boldsymbol{\phi}(\mathbf{x}')^\mathsf{T}] = \mathsf{T}[\boldsymbol{\phi}].$$

Then we can compute the approximate Koopman matrix as the ordinary least square regression, which is equivalent to the EDMD:

$$\hat{K}_\phi^{\mathsf{ols}} = (\mathsf{M}_\pi[\boldsymbol{\phi}])^{-1}\mathsf{T}[\boldsymbol{\phi}] = (\mathsf{M}_\pi[\boldsymbol{\phi}])^{-1}\mathsf{M}_\pi[\boldsymbol{\phi}, \boldsymbol{\alpha}]\mathsf{M}_\pi[\boldsymbol{\phi}, \boldsymbol{\beta}]^\mathsf{T}.$$

**Experimental Setup.** Setting $N = 20$, we generated a random trajectory of length $16384$ to construct the pair data. We used a multi-layer perceptron (MLP) with hidden units `64-128-64` and leaky ReLU activation for left and right parametric singular functions. We evaluated VAMPnet-1 (blue), DPNet (green), DPNet-relaxed (green with marker x), and LoRA (red) across varying batch sizes $B \in \{256, 1024, 8192\}$ and learning rates $\mathsf{lr} \in \{10^{-3}, 10^{-4}\}$ using the Adam optimizer; see Figure 4. Each configuration was trained for 500 epochs. We conducted five independent runs per configuration with different random seeds and report the average values with standard deviation. We set the number of modes to $k = 20$.

**Results.** To evaluate the quality of the learned singular basis, we first estimated the singular values $\hat{\sigma}_i$ via CCA. Since the ground-truth spectrum is known in this setting, we computed the relative error in estimating the squared singular values, defined as $\frac{|\sigma_i^2 - \hat{\sigma}_i^2|}{\sigma_i^2}$ for each $i \in [k]$. As shown in the first row of Figure 4, LoRA consistently outperforms the other methods in capturing the singular subspaces according to this metric, except in the configuration with a large batch size $B = 8192$ and a small learning rate $10^{-4}$. We also note that the performance of VAMPnet-1 is highly sensitive to the batch size, whereas DPNets exhibit relatively robust performance across settings.

Next, we assessed the quality of the estimated eigenvalues. Given the CCA-aligned basis $\mathbf{b} \in \{\tilde{\boldsymbol{\phi}}, \tilde{\boldsymbol{\psi}}\}$, we computed the Koopman matrix via EDMD using the first $i$ basis vectors $\mathbf{b}_{1:i}$ and extracted the corresponding eigenvalues $(\hat{\lambda}_j)_{j=1}^i$. Since the true system has three dominant eigenvalues $\lambda_1, \lambda_2, \lambda_3$, we evaluated the estimation quality using the directed Hausdorff distance $\max_{i' \in [i]} \min_{j \in [3]} |\hat{\lambda}_{i'} - \lambda_j|$, following Kostic et al. [16]. The results are presented in the second row of Figure 4.

As expected, increasing the number of singular functions improves the estimation quality across all methods. Kostic et al. [16] reported a baseline value of $0.06_{\pm 0.05}$ achieved by DPNet-relaxed (indicated by the gray, dashed horizontal line), and nearly all our configurations outperform this baseline. We observe that DPNet-relaxed and LoRA yield comparable performance, with DPNet-relaxed occasionally achieving the best results. We attribute the discrepancy between singular subspace quality and eigenvalue estimation performance to the non-normal nature of the underlying dynamics.

### G.2 Ordered MNIST

Following the setup of [16], we generated two independent trajectories of length 1000. We used one trajectory for training and the other for evaluation. We used Adam optimizer with learning rate $10^{-3}$, batch size 64, for 100 epochs. The convolutional neural network we used is same as [16], namely, `Conv2d[16]`→`ReLU`→`MaxPool[2]`→`Conv2d[32]`→`ReLU`→`MaxPool[2]`→`Linear[10]`. We set the metric deformation loss coefficient to be 1 as suggested for DPNet and DPNet-relaxed.

---

[9]We note that in the original implementation of Kostic et al. [16], the singular values were computed as if $\frac{p(x'|x)}{\rho_1(x')} = \boldsymbol{\alpha}(x)^\mathsf{T}\breve{\boldsymbol{\beta}}(x')$, which led to wrong singular values.

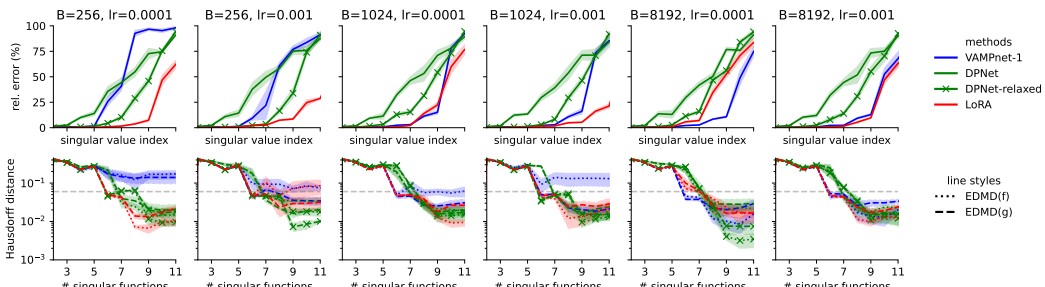

Figure 4: Summary of the noisy logistic map experiment. The first row reports relative error in singular values, and the second row the directed Hausdorff distance of the estimated eigenvalues to the three most dominant underlying eigenvalues. The shaded area indicates $\pm 0.5$ standard deviation.

### G.3 Langevin Dynamics

Consider the following stochastic differential equation (SDE):

$$d\mathbf{X}_t = \mathsf{A}(\mathbf{X}_t)dt + \mathsf{B}(\mathbf{X}_t)d\mathbf{W}_t, \tag{17}$$

where $\mathsf{A}\colon \mathbb{R}^d \to \mathbb{R}^d$ is a drift term, $\mathsf{B}\colon \mathbb{R}^d \to \mathbb{R}^{d\times m}$ is the diffusion term, and $\mathbf{W}_t$ is a $m$-dimensional Wiener process. Since the diffusion process can be modeled as a reversible stochastic differential equation (SDE) with respect to its stationary distribution $\pi$, the associated generator becomes self-adjoint within the Hilbert space endowed with the inner product $\langle \cdot, \cdot \rangle_\pi$ [50]. For the Itô SDE in Eq. (17), we can write the action of the generator $\mathcal{L}$ (also called the Kolmogorov backward operator, or the Itô derivative) of the diffusion on a function $f\colon \mathbb{R}^d \to \mathbb{R}$ can be written as, by the Itô formula [50],

$$(\mathcal{L}f)(\mathbf{x}) \triangleq \nabla_{\mathbf{x}} f(\mathbf{x})^\intercal \mathsf{A}(\mathbf{x}) + \frac{1}{2}\operatorname{tr}\Big(\mathsf{B}(\mathbf{x})^\intercal \nabla_{\mathbf{x}}^2 f(\mathbf{x})\mathsf{B}(\mathbf{x})\Big).$$

**Special 1D Example.** Consider a simple 1D case where

$$A(x) = -\frac{1}{\gamma}U'(x) \quad \text{and} \quad B(x) = \sqrt{\frac{2k_B T}{\gamma}}.$$

In this case, we can write

$$(\mathcal{L}f)(x) = \frac{1}{\gamma}\Big(-U'(x)f'(x) + k_B T f''(x)\Big).$$

For simplicity, we set $k_B T = 1$ and $\gamma = 0.1$.

We use the *Schwantes potential* $U(x)$ defined as

$$U(x) = 4\left(x^8 + 0.8e^{-80x^2} + 0.2e^{-80(x-0.5)^2} + 0.5e^{-40(x+0.5)^2}\right),$$

whose derivative is given as

$$U'(x) = 32x^7 - 512xe^{-80x^2} - 128(x-0.5)e^{-80(x-0.5)^2} - 160(x+0.5)e^{-40(x+0.5)^2}.$$

### G.4 Chignolin Molecular Dynamics

This section provides the detail of our experimental setup including the data, as well as a supplementary analysis, which demonstrates the stabilization effect of the nesting techniques during training.

**Experimental Setup.** Our work utilizes a publicly available trajectory dataset of the classic chignolin peptide (sequence: GYDPETGTWG, PDB: 1UAO) simulated at 300 K, provided by [25]. In contrast, contemporary studies [3, 16] analyze simulation data at 340K of the stabilized CLN025 variant (sequence: YDPETGTWY, PDB: 5AWL; [9]) from [20], which is not publicly available. To maintain maximal consistency with the prior works, we selected trajectories from this dataset generated using the CHARM22* force field and TIP3P solvent, but with a halved sampling interval of 100 ps. Moreover, unlike the non-public data created from a single trajectory, the full dataset contains approximately 300,000 snapshots from 34 trajectories, initiated from either folded or unfolded states.

Due to the short trajectory lengths, transition events are relatively rare in this dataset [25]. Therefore, naive random splitting may lead to disjoint ensembles, for example, a training set lacking transition events while the test set contains them. To mitigate this distribution shift and decouple data sampling variance from optimization variance, we employed a distribution-matching split strategy. Specifically, we generated 200 candidate stratified splits and selected the partition that minimizes the 1-Wasserstein distance between the training and test set distributions of the radius of gyration, computed using the 10 $C\alpha$ atoms. To ensure balanced initial conditions, we further constrained the split such that the 17 folded-initiated and 17 unfolded-initiated trajectories were each divided into 13 training and 4 testing trajectories.

After fixing the train-test split, all experiments were performed with five independent model training runs using this fixed dataset. This allows us to evaluate numerical stability and convergence robustness of the learning algorithms.

We adopted the SchNet-based architecture and hyperparameters from [16]. Therefore, we trained the models for 100 epochs with the Adam optimizer whose learning rate is $10^{-3}$, but doubled the batch size to 384 to maximize GPU memory utilization. We also used $\gamma = 0.01$ for the regularization coefficient of DPNet-relaxed. The SchNet architecture used in our experiments is summarized in Table 4.

Table 4: Architecture and hyperparameter configuration of the `SchNetModel` used in our experiments. The model processes atomic structures to produce fixed-dimensional feature vectors. Key hyperparameters were adopted from [16] for consistency with prior work.

| Component / Layer | Key Parameter | Value in Experiments |
|---|---|---|
| **Input** | Dictionary of atomic properties (positions, atomic numbers, etc.) | |
| `PairwiseDistances` | Cutoff radius (`cutoff`) | 6.0 Å |
| `GaussianRBF` | Number of radial basis functions (`n_rbf`) | 20 |
| `SchNet Blocks` | Feature dimension (`n_atom_basis`) | 64 |
| | Number of interaction blocks (`n_interactions`) | 3 |
| | Cutoff function | `CosineCutoff` |
| Output Layer (`nn.Sequential`) | | |
| 1. Linear Layer | Output dimension (`n_feature_modes`) | 15 (for k=16 modes with centering) |
| 2. `BatchNorm1d` | Use batch normalization (`use_batchnorm`) | True |
| **Output** | Per-atom feature tensor of shape (Total Atoms, `n_feature_modes`) | |

Since we utilized batch normalization in the output layer, the learned feature vectors are constrained to have zero mean over the batch. This effectively enforces orthogonality to the constant eigenfunction. To explicitly account for this structure, we applied the *centering* strategy (Section 2.3) as a default for all algorithms, modeling the constant mode separately. Consequently, the output dimension of the SchNet module was set to $k - 1 = 15$. This differs from the 16 dimensions used by Kostic et al. [16], as we incorporate the fixed constant mode as an additional, separate feature.

**Implementation of VAMP-E Score.** To evaluate the performance of the learned bases, we compute the VAMP-E score proposed by [48] to evaluate the generalization capability of the learned Koopman operator. Unlike the VAMP-$r$ scores which measure the canonical correlations within a specific

dataset, the VAMP-E score estimates the approximation error of the learned model with respect to the true Koopman operator in the Hilbert-Schmidt norm [48].

Specifically, maximizing the VAMP-E score is equivalent to minimizing the decomposition error $\|\hat{\mathcal{K}}-\mathcal{K}\|_{\mathrm{HS}}^2$ on unseen data. The squared error can be expanded as $\|\hat{\mathcal{K}}-\mathcal{K}\|_{\mathrm{HS}}^2 = \|\hat{\mathcal{K}}\|_{\mathrm{HS}}^2 - 2\langle\hat{\mathcal{K}},\mathcal{K}\rangle_{\mathrm{HS}} + \|\mathcal{K}\|_{\mathrm{HS}}^2$. Since $\|\mathcal{K}\|_{\mathrm{HS}}^2$ is an unknown constant dependent only on the system, minimizing the error is equivalent to maximizing the score $\mathcal{R}_E \triangleq 2\langle\hat{\mathcal{K}},\mathcal{K}\rangle_{\mathrm{HS}} - \|\hat{\mathcal{K}}\|_{\mathrm{HS}}^2$. Here, the first term measures the consistency between the learned model and the actual test dynamics, while the second term serves as a regularization term penalizing the model complexity on the test distribution.

In our implementation, we calculate the VAMP-E score using a cross-validation strategy where the model parameters are fixed based on the training data, and the score is evaluated on the test moments. Let $\hat{\mathsf{M}}_{\rho_0^{\mathrm{test}}}[\mathbf{f}]$, $\hat{\mathsf{M}}_{\rho_1^{\mathrm{test}}}[\mathbf{g}]$ and $\hat{\mathsf{T}}_{\mathrm{test}}[\mathbf{f},\mathbf{g}]$ denote the empirical second-moment matrices calculated from the test set. We first project these moments onto the singular subspace learned from the training data. Let $\mathsf{W}_0 \triangleq (\mathsf{M}_{\rho_0^{\mathrm{train}}}[\mathbf{f}])^{-1/2}$ and $\mathsf{W}_1 \triangleq (\mathsf{M}_{\rho_1^{\mathrm{train}}}[\mathbf{g}])^{-1/2}$ be the whitening matrices. Furthermore, let $\mathsf{U},\mathsf{V}$ and $\Sigma$ be the singular vectors and the diagonal matrix of singular values, respectively, obtained from CCA with respect to the training data. The projected test moments are defined as:

$$\mathsf{C}_{00} \triangleq \mathsf{U}^\top \mathsf{W}_0 \hat{\mathsf{M}}_{\rho_0^{\mathrm{test}}}[\mathbf{f}]\mathsf{W}_0\mathsf{U} \;=\; \Sigma^{-1/2}\hat{\mathsf{M}}_{\rho_0^{\mathrm{test}}}[\tilde{\phi}]\Sigma^{-1/2},$$
$$\mathsf{C}_{01} \triangleq \mathsf{U}^\top \mathsf{W}_0 \hat{\mathsf{T}}_{\mathrm{test}}[\mathbf{f},\mathbf{g}]\mathsf{W}_1\mathsf{V} = \Sigma^{-1/2}\hat{\mathsf{T}}_{\mathrm{test}}[\tilde{\phi},\tilde{\psi}]\Sigma^{-1/2},$$
$$\mathsf{C}_{11} \triangleq \mathsf{V}^\top \mathsf{W}_1 \hat{\mathsf{M}}_{\rho_1^{\mathrm{test}}}[\mathbf{g}]\mathsf{W}_1\mathsf{V} = \Sigma^{-1/2}\hat{\mathsf{M}}_{\rho_1^{\mathrm{test}}}[\tilde{\psi}]\Sigma^{-1/2}.$$

Using the singular value matrix $\Sigma$ computed from CCA with training data, the VAMP-E score is then defined as:

$$\mathcal{R}_E \triangleq 2\underbrace{\mathrm{tr}(\Sigma\mathsf{C}_{01})}_{\approx\langle\hat{\mathcal{K}},\mathcal{K}\rangle_{\mathrm{HS}}} - \underbrace{\mathrm{tr}(\Sigma\mathsf{C}_{00}\Sigma\mathsf{C}_{11})}_{\approx\|\hat{\mathcal{K}}\|_{\mathrm{HS}}^2} = 2\,\mathrm{tr}(\hat{\mathsf{T}}_{\mathrm{test}}[\tilde{\phi},\tilde{\psi}]) - \mathrm{tr}(\hat{\mathsf{M}}_{\rho_0^{\mathrm{test}}}[\tilde{\phi}]\hat{\mathsf{M}}_{\rho_1^{\mathrm{test}}}[\tilde{\psi}]). \qquad (18)$$

Intuitively, this formulation penalizes the model if the singular values $\Sigma$ learned from training are inconsistent with the correlations observed in the test data ($\mathsf{C}_{01}$), or if the learned basis functions lose orthonormality on the test set ($\mathsf{C}_{00},\mathsf{C}_{11} \neq \mathbf{I}$). Therefore, VAMP-E serves as a rigorous metric for validating both the learned singular subspaces and the estimated timescales [24].

**On the Stabilization Effect of Nesting Techniques.**

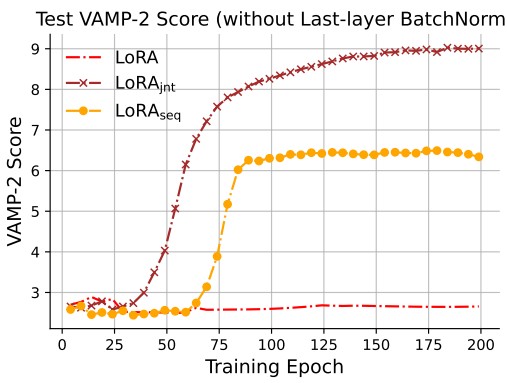

Figure 5: Test VAMP-2 score on the chignolin dataset for LoRA variants trained without the final batch normalization layer. While the standard LoRA objective without nesting fails to converge, both nesting techniques LoRA$_{\mathrm{jnt}}$ and LoRA$_{\mathrm{seq}}$ achieve high scores, suggesting improved optimization behavior in this challenging setting.

As an additional study, we report a further benefit of nesting techniques during training. To create a more challenging learning setup, we remove the final batch normalization layer from the SchNet architecture and train for 200 epochs. Figure 5 shows that the standard LoRA model fails to learn meaningful dynamics, as indicated by its low and stagnant VAMP-2 score. In contrast, both joint and sequential nesting exhibit stable training behavior and converge to high-scoring solutions. This

suggests that nesting techniques are not merely post hoc alignment mechanisms, but also provide stronger optimization signals in this challenging setting. We leave a more in-depth analysis of this empirical observation for future work.

## G.5 Computing Resources

All experiments were conducted on a server equipped with two Intel(R) Xeon(R) Gold 5220R CPUs, about 500 GiB of total RAM, four NVIDIA A5000 GPUs, and a 1TB NVMe SSD for storage. The chignolin molecular dynamics experiment was the most computationally intensive; each training run required 2-3 hours on two NVIDIA A5000 GPUs. Considering multiple runs across 5 data splits (Appendix G.4), the total computational effort for the reported chignolin experiments was equivalent to approximately 2-3 days. The other experiments (noisy logistic map, ordered MNIST, Langevin dynamics) were less demanding, completing in approximately 1 hour, typically using one A5000 GPU. All experiments utilized PyTorch; the chignolin simulations additionally employed SchNetPack 2.1.1 [35]. The total compute for the entire research project, including preliminary experiments, is estimated at approximately 3 days of operational time on this hardware.

