# OpenReview forum: "Efficient Parametric SVD of Koopman Operator for Stochastic Dynamical Systems"
_NeurIPS.cc/2025/Conference — NeurIPS 2025 poster_

### Official Review · Reviewer_acob · 2025-06-17

**Clarity:** 2
**Significance:** 3
**Originality:** 3
**Rating:** 4
**Confidence:** 3

**Summary:**

The authors propose a row rank approximation method to approximate Koopman operators. The simplicity of the algorithm makes it easy for the method to integrate with modern deep learning pipelInes by eliminating the need for unstable linear
algebraic operations.

**Questions:**

The singular vectors should be orthogonal each other. How do you guarantee the orthogonality of the vectors $f_1,\ldots, f_k$?

**Ethical Concerns:**

["NO or VERY MINOR ethics concerns only"]

**Final Justification:**

The authors addressed my comments properly. Thus, I updated my score.

**Limitations:**

Investigating what type of neural networks can be used for constructing $\mathbf{f}$ and $\mathbf{g}$ would be interesting.

**Paper Formatting Concerns:**

N / A

**Quality:**

2

**Strengths And Weaknesses:**

Finding good low-dimensional space for representing Koopman operators is a nonobvious and important problem. This paper try to tackle this problem, and the topic is related to the community.

There are existing methods that aim to learn low-dimensional spaces for representing Koopman operators. The advantage of learning a singular space is not clear for me. Could you clarify the advantage of learning a singular space instead of learning general linear spaces for low-dimensional approximation of Koopman operators?

Minor comments:
- Line 49: what do you mean by the identity $x_0=x_0$?
- Line 102: it seems that the limit is taken in the topology of the function space where the Koopman operator is defined, not the strong operator topology for operators.
- Line 115: "for even for " should be "even for"?

---

> ### Author Rebuttal · Authors · 2025-07-31
>
> We appreciate the effort and time in reviewing our manuscript. We address the reviewer's questions individually as follows.
>
> ---
>
> 1. **On the Advantage of Learning Singular Spaces**
> The reviewer asks why it is preferable to learn a singular subspace instead of a general linear space. While we are not sure if we understand the reviewer’s definition of a "*general linear space,*" the choice of learning singular subspaces for linear operator regression can be justified as follows.
>
>     For a general stochastic dynamical system, the corresponding Koopman operator may be non-normal, for which there is no spectral theorem. In such cases, there is no systematic way to recover the eigenfunctions of the Koopman operator that dominate the dynamics. Due to the inherent difficulty, prior works (i.e., VAMPnet and DPNet) instead formulate the problem as first learning the singular subspaces, which can be done systematically even for non-normal operators as long as they are compact, and then performing an operator regression step on top of them. For the special case of normal operators, EVD can be done by SVD, and thus SVD is fully justified. All in all, the choice of learning the singular subspaces is justified by the well-definedness of SVD for compact operators and the available computational techniques for computing SVD.
>
> 2. **On Guaranteeing Orthogonality of Singular Vectors**
>     Our method guarantees orthogonality in a two-step process:
>
>    1. **Implicit Orthogonality via Loss Minimization**: We implicitly enforce orthogonality by minimizing the objective function. Our LoRA loss function $-2\\mathrm{tr}(\\mathsf{T}\[\\mathbf{f}, \\mathbf{g}\]) \+ \\mathrm{tr}(\\mathsf{M}\[\\mathbf{f}\] \\mathsf{M}\[\\mathbf{g}\])$ is derived from minimizing $\\|\\mathcal{K} \- \\sum\_i f\_i \\otimes g\_i\\|\_{\\text{HS}}^2$. The global minimum of this loss is achieved when the learned functions $\\{f\_i\\}$ and $\\{g\_i\\}$ form bases for the true top-K singular subspaces. While the neural network outputs $\\mathbf{f}(x)$ and $\\mathbf{g}(x)$ may not be exactly orthogonal during training, the optimization process **drives them towards forming an orthogonal basis** for the target singular subspaces, as this is the configuration that minimizes the approximation error.
>    2. **Explicit Orthonormalization for Inference**: For downstream tasks (like eigen-analysis or prediction), we do not assume the raw neural network outputs are perfectly orthogonal. Instead, as described in **Approach 1 (CCA \+ LoRA, Sec 3.2)**, we perform a post-hoc canonical correlation analysis (CCA). This step explicitly computes the whitened bases $\\tilde{\\mathbf{f}}(x)$ and $\\tilde{\\mathbf{g}}(x')$ and the rotation matrices (from the SVD of $\\mathsf{T}\[\\tilde{\\mathbf{f}},\\tilde{\\mathbf{g}}\]$) to yield the final, strictly orthonormal singular functions $\\phi(x)$ and $\\psi(x')$.
>
>     In summary, our framework first learns a useful, near-orthogonal basis via stable loss minimization and then refines it into a strictly orthonormal basis for reliable downstream analysis.
>
> 3\. **Responses to Minor Comments**
>
> We appreciate the reviewer's comments on typos and clarifications. We will carefully revise our manuscript, incorporating these points.
>
> * **Line 49**: We apologize for the confusion. We meant the posterior mean of $g(\\mathbf{x}\_t)$ given a state $\\mathbf{x}\_0$ at time $t=0$.
> * **Line 102**: We meant the strong operator topology defined on a function space like Hilbert space $\\mathcal{H}$. That is, locally convex topology induced by seminorms $\\mathcal{T} \\mapsto \\|\\mathcal{T}(g)\\|\_{\\mathcal{H}}$ ($g\\in\\mathcal{H}$). In this topology, $\\mathcal{T}\_n \\rightarrow \\mathcal{T}$ iff $\\| \\mathcal{T}\_n(g) \- \\mathcal{T}(g) \\|\_{\\mathcal{H}} \\rightarrow 0$ for all $g \\in \\mathcal{H}$.
> * **Line 115**: Thank you for catching the typo. It should be "even for".
>
> ---
>
> We hope these clarifications address the reviewer's concerns, and if they do, we would be grateful if the reviewer would consider updating the score to reflect the contribution to the community.

---

> > ### Comment · Reviewer_acob · 2025-08-05
> >
> > Thank you for the clarification. I think my comments are properly addressed. I updated my score accordingly.

---

> > > ### Author Response · Authors · 2025-08-06
> > >
> > > Thank you for taking the time to read our rebuttal and update your evaluation. We are glad to hear that your concerns have been addressed. We truly appreciate your feedback and will revise the manuscript accordingly to prevent any potential confusion.

---

### Official Review · Reviewer_ZVY9 · 2025-06-30

**Clarity:** 2
**Significance:** 3
**Originality:** 3
**Rating:** 4
**Confidence:** 3

**Summary:**

This manuscript presents a novel method, termed LoRA, for learning the top-k singular functions of the Koopman operator for stochastic dynamical systems. The authors identify a critical issue in existing deep learning-based approaches like VAMPnet and DPNet: their reliance on numerically unstable operations such as matrix inversion and singular value decomposition within the training loop, which can lead to biased gradients and optimization challenges. To address this, the proposed LoRA method is based on directly minimizing the Hilbert-Schmidt norm of the low-rank approximation error. This formulation results in a loss function that is a simple polynomial of empirical second-moment matrices, thereby avoiding unstable computations during optimization. The method's efficacy and stability are demonstrated through a series of experiments on synthetic and molecular dynamics datasets, where it is shown to outperform baseline methods.

**Questions:**

Based on the analysis, we pose the following questions to the authors. First, in Theorem C.3 of Appendix C.2, there is a clear discrepancy. The theorem states that for a left eigenfunction $\zeta$, the relationship is $\mathcal{K}^{*}\zeta=\lambda\zeta$. However, the final step of the proof on line 832 derives the result as $\overline{\lambda}\zeta(x')$, where $\overline{\lambda}$ is the complex conjugate of $\lambda$. This conclusion from the proof, $\mathcal{K}^{*}\zeta=\overline{\lambda}\zeta$, is mathematically correct for adjoint operators but contradicts the theorem's statement. We ask the authors to clarify and correct this imprecision.

Second, the authors acknowledge in Appendix G.4.4 that the short lag time used in the Chignolin analysis "inherently biases the model towards resolving faster dynamic events." This raises a critical question about the method's general applicability: for a new system, how should a practitioner select an appropriate lag time to ensure the discovery of the most critical, slow dynamics, rather than being misled by parameter choices into focusing on faster, less relevant modes? Does this imply that the successful application of LoRA requires extensive prior knowledge or tedious sensitivity analyses?

Third, while the stability of LoRA is well-demonstrated, the experimental validation relies significantly on systems that could be considered "toy" or methodologically "tricky." The Noisy Logistic Map, for instance, is a 1D system chosen specifically because its finite-rank kernel is analytically tractable. Similarly, the Ordered MNIST experiment uses a synthetic process with simple, deterministic dynamics layered on top of image data. While valuable for controlled comparisons, how confident are the authors that the strong performance on these simplified or contrived benchmarks will translate to more complex, high-dimensional, and chaotic real-world systems where the underlying dynamics are not low-rank or neatly structured?

**Ethical Concerns:**

["NO or VERY MINOR ethics concerns only"]

**Limitations:**

The most fundamental limitation of this research is that, as a data-driven method, the quality of the learned dynamical model is entirely beholden to the quality and characteristics of the input data. The Chignolin experiment serves as a primary example: if the available data, due to factors like trajectory length or sampling frequency, does not sufficiently contain the signal of the slowest physical processes (e.g., microsecond-scale global folding), the method cannot recover them, regardless of its mathematical optimality or algorithmic stability. Therefore, the method cannot guarantee the discovery of the absolute slowest physical timescale in all cases; rather, it finds the most prominent observable slow process within the given data. This data-dependency defines the principal boundary for the method's applicability when applied to new systems.

**Paper Formatting Concerns:**

No major formatting issues in this paper.

**Quality:**

2

**Strengths And Weaknesses:**

The primary strength of this work lies in its clear identification of a key technical bottleneck in existing Koopman analysis methods—namely, numerical instability during optimization—and its proposal of a mathematically rigorous and algorithmically elegant solution. The LoRA objective function is well-motivated, and its stable polynomial form represents a significant advantage, facilitating straightforward implementation and optimization within modern deep learning frameworks. The derivation provided in the appendix confirms its theoretical soundness. The experimental results, particularly the successful application to the complex Chignolin molecular dynamics system where baseline methods reportedly failed to converge, compellingly demonstrate the superior stability and practical utility of the proposed algorithm.

However, the rigor and persuasiveness of the work are undermined by several significant flaws in its reporting and execution. Firstly, the reporting on the crucial Chignolin experiment contains multiple errors in the main manuscript, including incorrect units, data splits, and table labels, which, while noted in an appendix errata, reflect a lack of rigor in the report itself. Secondly, the work’s central conclusion relies heavily on a post-hoc physical justification in Appendix G.4.3 to bridge the gap between their experimental data and a reference value from a different temperature; this justification is qualitative and lacks quantitative support, making it appear speculative. Most critically, the authors admit in Appendix G.4.4 that their choice of a short lag time "inherently biases the model towards resolving faster dynamic events," which is a concession of a systemic flaw in the experimental design that makes it unsuitable for validating the theoretical goal of finding the system's slowest physical processes.

---

> ### Author Rebuttal · Authors · 2025-07-31
>
> We thank the reviewer for their thorough review, which has helped us strengthen our manuscript. We address the reviewer's concerns individually below.
>
> ---
>
> 1. **On the Chignolin Experiment: Lag Time, Reference Timescale, and LoRA's Robustness**
>
>     1. `On the Choice of Lag Time`: The reviewer points out that `the authors acknowledge in Appendix G.4.4 that the short lag time used in the Chignolin analysis "inherently biases the model towards resolving faster dynamic events."`, and raises a valid concern regarding how lag time should be chosen and whether this necessitates sensitivity analysis or extensive prior knowledge. We apologize for any confusion this statement may have caused, and we would like to clarify its intended meaning below.
>
>         In an MD analysis setup, there exists two important design parameters:
>          - **(1) Lag time**, which is the resolution to emulate/discretize the continuous time dynamics. If the lag time gets longer, the discretized trajectories will only contain very low-frequency dynamics. As the lag time gets shorter, the trajectories will capture more refined, high-frequency information of the underlying dynamics, but it may capture more stochastic noise which thus could make the analysis more difficult.
>          - **(2) Trajectory length**, which is the length of the simulated trajectory. As a trajectory gets longer, the experiment setup aims to capture longer dynamics, and vice versa.
>
>         Our experiment analyzes rather short-length trajectories in a short lag time regime (100 ps).
>         Some prior works in the literature analyze the trajectories of sweeping the lag time for a complete understanding of the data, see, e.g., \[Fig. 6, 22\]. In the current work, we aim to demonstrate whether the considered methods can reliably fit to the high-dimensional trajectories in the challenging short lag time regime.
>
>         In this setup, we show that **LoRA can reliably extract the dominant, slow dynamics from high-dimensional trajectories, even in scenarios where DPNet-Relaxed only achieves strictly subpar performance in terms of the both training/test VAMP-2 score, while all other methods completely fail to converge**.
>
>     2. `Regarding the Timescale Analysis`
>         The reviewer raises a concern on the validity of the timescale analysis of the Chignolin experiment. We agree with the reviewer’s sentiment on the issue, and here we clarify the purpose and validity of such analysis. First of all, we emphasize that the concerned molecular dynamics is normal, and thus accurate estimation of singular functions with a high test VAMP-2 score directly implies accurate learning of the dynamics. Therefore, the subsequently high VAMP-2 score of LoRA variants against the baseline methods should be the main experimental takeaway of the experiment. Given this, the timescale analysis, a standard post-hoc analysis in this field \[19,22\], is to provide a physical relevance of the answer from the computational method. Our close estimate to the known reference value mildly suggests that our computational result is on the track, but this does not indicate any scientific conclusion. Moreover, as the reviewer rightfully noted, we have some additional discussions in the appendix to further compensate for the different configurations from the reference.We acknowledge that this physical discussion may distract our main message. We thus will tone down the current emphasis on the timescale analysis, but rather emphasize the good fit of LoRA to the underlying data.
>
> 2. **On Reporting Rigor and Theorem C.3**
>
>     We sincerely thank the reviewer for their meticulous reading.
>
>     * **Reporting Errors**: We agree that the errors in the main text (e.g., units, data splits) were oversights. We have corrected all of them to ensure the manuscript is accurate.
>     * **Regarding Theorem C.3**: After careful reexamination, we have found no inconsistency between the statement and proof in Theorem C.3. Namely, we claim that $\\mathcal{K}^{\*}\\zeta=\\overline{\\lambda}\\zeta$ in the statement, and prove it subsequently. If there is anything we have missed, please let us know.
>
> 3. **On Performance in More Complex, Real-World Systems**
>
>     - While our benchmarks were chosen for their controlled properties, they were designed to test the very failure modes we aimed to solve. Our experiments validate the core advantage of our method that are crucial for real-world application with complex dynamics. The **Noisy Logistic Map** and **Ordered MNIST** were selected because they exhibit non-normal or structured dynamics where optimization stability is critical. The Chignolin system, while smaller than real-world proteins, represents a real and challenging molecular dynamics problem.
>
>     - Regarding its applicability in more challenging real-world setups, we believe that the normal dynamics can be relatively easily handled by the LoRA framework even if it is not low-rank, as we can increase the number of modes to fully capture the dynamics. We conjecture that dealing with non-normal dynamics such as chaotic systems will remain as great challenges, as accurate estimation of non-normal dynamics in the sense of accurate top-K SVD may fail to capture some dominant modes in its eigen analysis. We note that this is an inherent limitation of available computational tools, not limited to the LoRA framework. That said, since we do not know how to directly compute eigenfunctions of non-normal dynamics, the top-K SVD framework will remain as an alternative solution that provides a systematic learning of dynamics, which could still provide some insights.
>
> 4. **Additional Experiments**
>
>     As we explained in our response to **Reviewer bFvN**, we conducted additional ablation study in the Chignolin experiment, by varying model dimensionalities and batch sizes to test the scalability and robustness of LoRA. The results  corroborate that LoRA's advantage translates to more complex, higher-dimensional settings. While further validation on larger systems is an important next step, we are confident that the fundamental stability we have demonstrated may allow the data-driven, deep-learning-based approach to be more widely applicable.

---

> > ### Author Response · Authors · 2025-08-06
> >
> > We thank you for your thorough and meticulous review. Your detailed feedback was invaluable, and we have posted a rebuttal that we hope addresses your main concerns.
> >
> > In particular, we have corrected the reporting errors you identified, clarified the rationale behind our experimental design for the Chignolin system (and will revise the emphasis as you suggested), and elaborated on Theorem C.3 in response to your comments.
> >
> > We wanted to politely check whether our rebuttal and planned revisions have sufficiently addressed your concerns. We would be grateful for any further questions or suggestions you may have.

---

> > ### Comment · Reviewer_ZVY9 · 2025-08-06
> >
> > We thank the authors for the detailed rebuttal. The response clarifies the method's stability advantages and rightly situates the work within the broader field's limitations. However, the justification for the key Chignolin experiment by reframing its goal from physical discovery to robustness testing weakens the paper's original and more impactful conclusions. So I will maintain the initial rating.

---

### Official Review · Reviewer_bFvN · 2025-07-01

**Clarity:** 3
**Significance:** 2
**Originality:** 1
**Rating:** 4
**Confidence:** 5

**Summary:**

This work studies a deep-learning–based framework for modeling dynamical systems by approximating the Koopman operator through a low-rank singular-value decomposition. Its principal contributions are evaluation of the Low-Rank (LoRa) contrastive loss in the context of the (stochastic) Koopman operator approximation, systematic comparisons with existing related methods, careful elaboration  of implementation details, and  and extensive experiments.

The chief advantage of LoRa is that its computation avoids the inversion of latent-space covariance matrices—objects that are often ill-conditioned during training. By avoiding these unstable inverses, LoRa produces unbiased estimates of the loss and its gradients, delivering more reliable optimization and superior performance compared with previous state-of-the-art losses.

**Questions:**

- **Figure 1 (top row):** Why does the singular-value error *increase* as the number of singular functions grows? Intuitively, adding more modes should *reduce* the approximation error. Please clarify the underlying reason.
- **Left vs. right singular spaces:** The paper compares the span of the left-right-singular-functions on downstream inference tasks, yet theory predicts that both should converge to the same Koopman-invariant subspace. What practical insight is gained from this comparison?.
- **Scope of the empirical study:** The work largely integrates results from prior literature and promises a thorough ablation of LoRa-based Koopman approximation. In practice, the only clear conclusion is LoRa’s superiority over DPNet and VampNet losses. To strengthen the empirical contribution, consider adding experiments that yield actionable insights on:
    - **Latent dimensionality:** How do all models scale as the latent space grows? We know that large latent spaces will lead to unstable DPNet and Vamp scores. Is LoRa based optimization unaffected by this?
    - **Orthonormal regularization:** Is orthonormal regularization influencial for numerical stability and/or convergence speed of the training process?
    - **Batch-size sensitivity:** How does batch size affect optimization stability of these methods?

**Ethical Concerns:**

["NO or VERY MINOR ethics concerns only"]

**Final Justification:**

While I do agree with the authors that this work is meaningful contribution to the community, and I believe that the paper is well written, I remain doubtful about clear acceptance of the presented contribution.

As I noted in the discussion with the authors. Previous works already demonstrated the role of the SVD of the transfer operator, as a specific case of the conditional expectation operator (CEO), in reliably learning dynamical systems, and demonstrated how one learns spectral features of CEO via MSE loss on its kernel. So, from my perspective the novelty just remains in **empirically testing existing approaches** on **already proposed  datasets** from dynamical systems. I don't find this strong enough contribution reflecting the score 5.

**Limitations:**

Section limitation is present, but the discussion is quite superficial and not discussing limitations per se, but rather plans for future work.

**Paper Formatting Concerns:**

OK

**Quality:**

3

**Strengths And Weaknesses:**

### __Strengths__

- The paper provides a solid review of related deep-learning approaches for stochastic Koopman operators and provides their comparisons.
- The paper is self contained, well written and accessible.
- The variants of the proposed algorithm appear to consistently beat state-of-the-art baselines.

### __Weakenesess__

- __Novelty:__ Although the paper cites [29] and [15], Section 3.1 should explicitly acknowledge that the LoRa loss was *first* introduced for low-rank conditional-expectation operator approximation in those works, while paper [14] proposed truncated SVD as the approach for representation learning for stochastic processes estimation. Additionally, I strongly suggest to place LoRa in the wider context of self-supervised contrastive learning and density ratio fitting by citing

    • Wang *et al.* (2022), “Spectral Representation Learning for Conditional Moment Models”

    • Sugiyama *et al.* (2012), “Density Ratio Estimation in Machine Learning”

- __Inconclusive empirical results:__ While the paper promises an experimental evaluation of design choices LoRa based low-rank approximation of the Koopman operator, many results appear inconclusive:
    - **Left vs. right singular functions**
    Figure 1 shows no clear performance gap, and Figure 2 reports no statistically significant difference. These results could be summarized in prose, omitting the corresponding curves to reduce visual clutter.

        The claim

        > “...we empirically found that using the right singular basis can sometimes yield better results.”
        >

        is not convincingly supported by the current data.

        Since theory predicts both subspaces converge to the same Koopman-invariant space once training stabilizes, this design choice may not warrant prominent discussion; consider moving it to the appendix.

    - **Joint vs. sequential nesting**
    The implementation of sequential nesting is unclear, given that all left/right functions are parameterized by the same encoder / delayed-encoder. Please clarify the procedure. Furthermore in the joint nesting, it is unclear how the weights are chosen given that the during learning the left-and-right data representations are not ordered by relevance of the singular space. It seems that if the nesting technique is going to be one of the main contributions of the paper, it warrants both better introduction and more conclusive results.

    - **Hyper-parameter study**
        In the synthetic experiment, Figure 1 shows no discernible trend for learning rate or batch size. It is hard for the reader to draw any conclusion from these plots, beyond the better performance of LoRa against baselines. Either try to pursue a deeper ablation study with more conclusive results and trends, or simplify this plot to avoid the clutter.

- __Minor issues:__  Font sizes in most plots are too small, compromising readability. Use the available whitespace to enlarge labels and legends. The x-axis of Figure 3 is unlabeled; specify the quantity and units. The paper does not report the scalar weighting the DPNet regularization term, nor whether this parameter was tuned.


__In summary, I believe that the paper is solid contribution with interesting empirical results, but my concern lies in clear and sufficient novelty w.r.t. the existing literature.__

---

> ### Author Rebuttal · Authors · 2025-07-31
>
> We appreciate the reviewer’s effort in reviewing our manuscript and providing constructive feedback.
> Below, we address each concern separately.
> (Note: Just to clarify our convention, we intended it to be LoR**A** for **Lo**w-**R**ank **A**pproximation.)
>
> ---
>
> 1. **On Novelty/Originality**
>
> We thank the reviewer’s comments on the literature (on the truncated SVD view in \[14\] and on self-supervised contrastive learning) and will incorporate these in our revision.
>
> However, we wish to remark that the idea of LoRA dates back to as early as Schmidt’s paper in 1907, which proposed it to characterize the top-$K$ eigenfunctions of integral operators. Given this, we emphasize that our key contribution is, in the field of data-driven learning of Koopman operators: (1) we first **identify** common, crucial flaws in the dominant optimization frameworks, namely VAMPnet and DPNet, and resolve them by **applying** the LoRA framework and concretely **demonstrating** its benefit by experiments.
>
> In this work, identifying the challenges in practical optimization, we show that LoRA, albeit a classical idea, can address the issues in the prior work in a principled yet simple way.  This stability is what allows us to reliably learn superior representations in high-dimensional settings, as demonstrated by our results. This is why, even on smaller benchmarks, our method consistently finds better solutions (e.g., higher VAMP-2/E scores and more accurate timescales in Chignolin), as it is hindered by neither the numerical instability nor gradient biased estimates during optimization that plague its predecessors. We will revise Sec 3.1 to explicitly state this distinction and place our work in the context of the suggested literature in a broader context.
>
> We would like to ask the reviewer to kindly reassess the “originality” of our work given this context.
>
> 2. **On the Scope of the Empirical Study: Scalability and Orthogonality**
>
> To provide more direct empirical evidence, we conducted new scalability and stress-test experiments on the Chignolin experiments, with data split for seed $0$. We compared the test VAMP-E score on our LoRA variants against DPNet-relaxed by increasing the number of modes (**N**) and the SchNet node feature dimension (**H**), while also reducing the batch size (**B**):
>
> |(B,N,H)|SeqLoRA|JntLoRA|LoRA|DPNet-relaxed|DPNet-relaxed-Ctr|
> |:-:|:-:|:-:|:-:|:-:|:-:|
> |(384, 64, 128)|31.61|29.89|29.51|-9.76${}^2$|5.20${}^1$|
> |(96, 64, 64)|25.83|23.82|25.51|4.75${}^1$|5.93${}^1$|
> |(384, 64, 64)|26.66|27.25|24.11|-1.97${}^2$|4.72${}^1$|
> |(384, 32, 128)|17.68|15.66|16.26|1.75|3.08|
> |(96, 32, 64)|17.19|15.64|17.36|4.03${}^1$|6.44|
> |(384, 32, 64)|16.69|17.02|15.79|4.36|4.91|
> |(384, 16, 64)|9.73|8.79|8.80|3.05|4.19|
> |(96, 16, 64)|10.22|8.80|8.52|2.74|4.40|
>
> (VAMP-E scores are calculated on overall test data, not distinguishing folded/unfolded initialization.)
> ${}^1$ During training, the VAMP-2 score exceeded the theoretical maximum (number of modes $N$). This strongly suggests numerical instability, suggesting a severe breakdown of orthogonality or a failure to learn diverse features.
> ${}^2$ The negative VAMP-E score implies that the approximation error $\\|\\mathcal{K} \- \\hat{\\mathcal{K}}\\|$ is larger than the (Hilbert-Schmidt) norm of the true operator $\\|\\mathcal{K}\\|$ itself, implying a poor approximation of the operator.
>
> Our experiments reveal that DPNet-relaxed completely fails when scaled to larger models or trained with smaller batch sizes. In direct contrast, our LoRA-based methods demonstrate stability across all settings, even at 4 times more modes. This experiment offers strong empirical proof that LoRA exhibits much more favorable scalability than the prior works.
>
> **Orthonormal Regularization**
>
> We answer the reviewer's question about the role of orthonormal regularization. Our LoRA method does not require explicit regularization to achieve stable convergence, which is a key advantage. The necessity of such regularization in prior methods stems from two interconnected issues:
>
> * **Theoretical Optimality**: As established by the Eckart-Young theorem, the best rank-K approximation of the Koopman operator is achieved by its top-K orthonormal singular functions. Therefore, learning an orthonormal basis is not just a heuristic but a core requirement for finding the optimal low-rank model.
> * **Practical Stability**: As the reviewer insightfully pointed out, maintaining this orthonormality is crucial for the stability of prior methods like DPNet. Their objectives rely on numerically unstable operations (e.g., matrix inversion). If the learned basis functions lose orthogonality and become collinear during training, these operations fail, leading to the training collapse we demonstrated in the Chignolin experiment.
>
> Our LoRA objective, by its design, sidesteps this issue. To provide a more direct evidence, we analyzed the orthonormality of the raw learned basis functions (before any post-hoc CCA) on the Chignolin test data (seed $0$). We measured the Frobenius distance between the normalized Gram matrix and the identity matrix, where the maximum is about $15.5$:
>
> |(B, N, H)|SeqLoRA|JntLoRA|LoRA|DPNet-relaxed|DPNet-relaxed-Ctr|
> |:---:|:---:|:---:|:---:|:---:|:---:|
> |(384, 16, 64)|0.38 / 0.53|0.30 / 0.26|0.21 / 0.25|6.13 / 6.17|5.37 / 5.32|
> |(96, 16, 64)|0.50 / 0.48|0.34 / 0.35|0.24 / 0.25|4.31 / 4.22|5.00 / 5.16|
>
> The results reveal an order-of-magnitude difference, showing that our LoRA variants naturally learn a nearly orthonormal basis directly from the optimization, while DPNet-relaxed fails to do so. This demonstrates that LoRA's stability is not just about preventing training failure but also about guiding the model toward a well-structured, theoretically optimal solution without explicit regularization.
>
> 3. **On the Clarity and Implementation of Nesting Techniques**
>
> We agree that some clarifications are warranted for nesting techniques, as we elaborate below. We will carefully revise our manuscript to strengthen our contribution.
>
> **Clarification on Sequential Nesting with Shared Encoders**
> As pointed out by the reviewer, we used “joint parameterization” for estimating singular functions throughout the paper.
>
> - The joint nesting is a principled technique that may guarantee to recover the desired solutions by finding the global optima regardless of the form of parameterization, as alluded to in line 200.
> - On the other hand, sequential nesting may not guarantee convergence for the case of joint parameterization, as it assumes independent parameter updates for different modes.
>
> In this paper, however, we tested and presented the performance of sequential nesting in this scenario as a working heuristic that can occasionally outperform joint nesting (Table 1). Therefore, similar to suggested in \[29\], we recommend using joint nesting in general for jointly parameterized eigenfunctions, but practitioners may consider applying sequential nesting as a heuristic with some caution of no guaranteed convergence. For the case of “separate parameterization” where each mode is parameterized by a separate network, sequential nesting may be preferred, as argued in \[29\]. We will carefully clarify these points in the manuscript.
>
> The detailed implementations for both versions of nesting are currently described in Appendix F. For sequential nesting, we achieve a sequential learning process by controlling the flow of the gradients. When training the $i$-th function pair ($f\_i,g\_i$), we use `torch.no_grad()` (referred to as `sg` in the pseudocode in Appendix F) to stop gradients from flowing back to the parameters that define the previously learned functions ($\\mathbf{f}\_{1:i-1}, \\mathbf{g}\_{1:i-1}$). This procedure effectively "freezes" the previously optimized subspaces, ensuring that the network focuses its updates only on learning the new, orthogonal mode. In our revision, we will make this implementation detail explicit in Section 3.1.
>
> **Weight Selection for Joint Nesting**
> In our experiments, we used uniform weights ($\\alpha\_i \= 1/k$ for all $i$), as we did not assume any prior knowledge about the relative importance of different singular modes. We will add this detail to the experimental setup section for full reproducibility.
>
> 4. **Responses to Specific Questions**
>
> * **Increase in Singular-Value Error**: As explained in line 284, the error in Figure 1 is the *relative* error, $(\\hat{\\sigma}\_i^2 \- \\sigma\_i^2)/\\sigma\_i^2$ for each index $i$. As the index $i$ increases, the true singular value $\\sigma\_i$ becomes smaller and less dominant. This causes the relative error to increase, even if the absolute error decreases. We will clarify this in the caption.
> * **Left vs. Right Singular Spaces**: We appreciate the reviewer's suggestion. We intended to provide a complete analysis for practitioners and show that either basis can be used for EDMD with similar performance, given that the literature only considered using the left eigenbasis. However, we agree this point can be de-emphasized to improve clarity. We will move the detailed comparison to the appendix in our revision.
> * **On Minor Issues and DPNet Regularization**: We agree that the plots can be revised to improve readability and will correct all visual elements (font sizes, axis labels) in the revised manuscript. Regarding the DPNet regularization parameter $\\gamma$, we followed the experimental setup of the DPNet paper \[14\] to ensure a fair and direct comparison with the reported results in the literature. We will clarify this in our experimental setup section.
> ---
> We hope our responses and planned revisions address the reviewer's concerns. If these clarifications adequately address the reviewer’s concerns, we would greatly appreciate it if the reviewer could consider updating the score to reflect what we believe are meaningful contributions to the community.

---

> ### Comment · Reviewer_bFvN · 2025-08-05
>
> I thank the authors for their replies. While the rebuttal successfully clarifies some aspects of their work, my main concern on the novelty remains.
>
> I do agree with the authors that this work is meaningful contribution to the community. However, [14] already demonstrated the role of the SVD of the transfer operator, as a specific case of the conditional expectation operator (CEO), in reliably learning dynamical systems, and several works demonstrated how one learns spectral features of CEO via MSE loss on its kernel [15, 29, 38] (but also [A, B]). So, contrasting current submission with the previous literature, from my perspective the novelty just remains in **empirically testing existing approaches** on **already proposed  datasets** from dynamical systems. Indeed, based on previous literature demonstrated results are as one would expect.
>
> Based on this, I am willing to increase my score to 4, remaining doubtful about clear acceptance of the presented contribution.
>
> **Ref.**
>
> [A] Izbicki, Rafael, and Ann B. Lee. "Converting high-dimensional regression to high-dimensional conditional density estimation." (2017): 2800-2831
>
> [B] Shimizu, Eiki, Kenji Fukumizu, and Dino Sejdinovic. "Neural-kernel conditional mean embeddings." arXiv preprint arXiv:2403.10859 (2024).

---

### Official Review · Reviewer_jeGy · 2025-07-07

**Clarity:** 3
**Significance:** 3
**Originality:** 3
**Rating:** 4
**Confidence:** 2

**Summary:**

This paper introduces a scalable and stable method for learning the top-k singular subspaces of the Koopman operator in stochastic dynamical systems using LoRA. Unlike prior methods such as VAMPnet and DPNet, which rely on backpropagation through numerically unstable operations like SVD or matrix inversion, LoRA formulates a variational objective entirely in terms of polynomial forms of moment matrices, enabling unbiased gradient estimation and efficient training using minibatches. The proposed method is validated on multiple benchmark systems, including synthetic chaotic maps, Langevin dynamics, molecular simulations (Chignolin), and Ordered MNIST, demonstrating superior performance in general.

**Questions:**

N/A

**Ethical Concerns:**

["NO or VERY MINOR ethics concerns only"]

**Final Justification:**

Due to the abovementioned reasons, my final rating is 4: borderline accept.

**Quality:**

3

**Strengths And Weaknesses:**

Strengths
- The paper is clearly structured, with a coherent flow from motivation, problem formulation, method, theory, and experiments.
- I’m not much familiar with the field, but the proposed approach’s motivation is convincing. e.g., challenges on VAMPnet and DPnet (e.g., unstable backpropagation, scalability) are clearly discussed.
- Notation is mostly consistent, and figures are helpful for understanding convergence and performance trends.

Weaknesses and Questions
- While LoRA seems not been applied to dynamic mode decomposition, it looks like a simple integration of existing LoRA method to existing problem to a certain extent.
- While the theoretical background and motivation are sound, some of the proof sketches and intuition for why LoRA performs better (especially under non-normal dynamics) could be clarified in the main text. Please correct me if it was presented in section 3.
- I agree that VAMPnet and DPnet would be more difficult to optimize. However, I'm not sure the experimental results effectively exhibit such limitations. In terms of rel. error in Figure 1, LoRA achieves the lowest error comparing others, but it’s not always true when evaluating Hausdorff (the lowest distance achieved by DPNet).
- Nesting (seq or jnt) results do not seem consistent. For example, LoRA_jnt is superior in Figure 2, but LoRA_seq is better in Table 1.
- Also, I expected results showing LoRA is significantly faster and scalable than VAMPnet and DPnet, but such experiments seem not considered.

As I’m only little familiar with the field, I’ll have to see other reviewers score to finalize mine.

---

> ### Author Rebuttal · Authors · 2025-07-31
>
> We appreciate the reviewer for their time and effort in reviewing our manuscript, providing thoughtful comments. We will address the reviewer's questions individually below. We will update our manuscript to improve the clarity, accompanying the feedback.
>
> ---
>
> 1. **On the Novelty of Our Work**
>
> We emphasize that our key contribution is to raise a key practical bottleneck in the dominant optimization frameworks for Koopman operator learning, and address it by **reconceptualizing the underlying variational objective**.
>
> As we detailed in the Item 1 of our response to Reviewer bFvN, prior methods like VAMPnet/DPNet, while theoretically sound, are built on objectives that are fundamentally incompatible with modern deep learning optimization due to their reliance on numerically unstable and statistically unfavorable operations such as SVD and inversion of empirical moment matrices. Our work is the first to identify this practical failure mode as a central problem and propose a principled solution via a direct low-rank approximation.
>
> 2. **On the Intuition Behind LoRA’s Superiority**
>
> Our method’s core advantage stems from its numerical stability, a crucial aspect that has been largely overlooked in the field of Koopman operator learning. As discussed in the first item, prior methods, including the recent VAMPnet and DPNet, suffer from operations like matrix inversion on empirical moment matrices. They may result in not only numerical instability for ill-conditioned empirical moments, but also biased gradient estimates, which may ultimately hinder their application to complex, large-scale dynamical systems.
>
> Our LoRA approach directly addresses this fundamental problem by reformulating the underlying variational principle of Koopman learning. Instead of relying on objectives that require unstable linear algebraic operations, we propose to directly minimize the operator norm of **low-rank approximation error** of the Koopman operator.
>
> The resulting objective function consists solely of polynomials of second-moment matrices, thereby eliminating the need for any unstable linear algebraic operations during training. Furthermore, we have theoretically validated the practicality of this objective function by proving its statistical consistency under finite-minibatch settings, as detailed in Appendix E.2. This combination of a stable formulation and theoretical guarantees provides the intuition for LoRA's superior performance and scalability.
>
> 3. **On Experimental Results: Stability vs. Performance Metrics**
>
> The primary goal of the experiments is to demonstrate that LoRA offers superior performance and scalability as promised. As the reviewer noted, in the Logistic map experiment, DPNet-relaxed shows superior performance in terms of the Hausdorff distance for estimated eigenvalues. This suggests that DPNet-type objectives may work as expected for small-scale systems.
>
> Our subsequent experiments reveal, however, that the DPNet(-relaxed) technique suffers to scale to high-dimensional setups. In the ordered MNIST experiment, where the environment is higher-dimensional, all three LoRA methods exhibit far better prediction compared to the baselines, including variants of DPNet. More critically, the continuous-time version of the DPNet objective also fails to accurately capture eigenfunctions of 1D Langevin dynamics. Ultimately, in the Chignolin experiments, VAMPnet and DPNet completely fail to converge, and DPNet-relaxed only achieves a far worse data fit compared to LoRA. In the item 1 of our response above, we further strengthen this scalability argument by showing that DPNet-relaxed becomes to break down (i.e., diverge or significantly underperform) on settings with larger models and smaller batch sizes.
>
> 4. **On Nesting Techniques**
>
> We acknowledge that the reviewer rightfully asked about a winning strategy among the two nesting techniques. As alluded to above (see item 1), we used “joint parameterization” throughout the paper.
>
> * On the one hand, the joint nesting is a principled technique that may guarantee to recover the desired solutions by finding the global optima regardless of the form of parameterization, as alluded to in line 200\.
> * On the other hand, sequential nesting may not guarantee convergence for the case of joint parameterization, as it assumes independent parameter updates for different modes.
>
> In this paper, however, we tested and presented the performance of sequential nesting in this scenario as “a working heuristic” that can occasionally outperform joint nesting (as shown in Table 1). Therefore, as suggested in \[29\], we recommend using joint nesting in general for jointly parameterized eigenfunctions, but practitioners may consider applying sequential nesting as a heuristic with some caution of no guaranteed convergence. For the case of separate parameterization where each mode is parameterized by a separate network, sequential nesting may be preferred, as argued in \[29\]. We will carefully clarify these points in the manuscript.
>
> Overall, we remark that our main claim about nesting is that nesting can help learning more than without it, leading to better performance in general (as in Figure 2). To provide direct evidence for this claim, we conducted a new, more challenging experiment on the Chignolin dataset. We deliberately made the optimization more difficult by removing the last batch normalization layer from the SchNet architecture and trained for 200 epochs. This setup tests the intrinsic stability of the gradient updates. The plots below, visualizing VAMP-2 score on test data, present a clear benefit of nesting. While the vanilla LoRA (without nesting) struggles to learn meaningful dynamics, in sharp contrast, both nesting (both joint and sequential) enable the model to converge to a high-quality solution, demonstrating that nesting acts as a critical regularizer and stabilizer.
>
> ```
>   10 +----------------------------------------+
>    9 |                                        |
>    8 |                           LoRA ******* |
>    7 |                                        |
>    6 |                                        |
>    4 |                                        |
>    3 |****                                    |
>    2 |    ************************************|
>    1 +----------------------------------------+
>      0   20  40  60  80  100 120 140 160 180 200
>   10 +----------------------------------------+
>    9 |                    ********************|
>    8 |             *******    JntLoRA ******* |
>    7 |           **                           |
>    6 |          *                             |
>    4 |         *                              |
>    3 |   *  ***                               |
>    2 |*** **                                  |
>    1 +----------------------------------------+
>      0   20  40  60  80  100 120 140 160 180 200
>   10 +----------------------------------------+
>    9 |                                        |
>    8 |                        SeqLoRA ******* |
>    7 |                 ***********************|
>    6 |               **                       |
>    4 |              *                         |
>    3 |            **                          |
>    2 |************                            |
>    1 +----------------------------------------+
>      0   20  40  60  80  100 120 140 160 180 200
> 			Epochs
> ```
>
> 5\. **On Stability and Scalability Compared to VAMPnet and DPNet**
> We agree that the advantages (stability and scalability) of LoRA can be better highlighted in our experiments.
>
> * The latter two experiments (Langevin dynamics and Chignoilin) more directly demonstrate the stability and scalability of LoRA. In the Langevin dynamics example, we were not able to train a model using the DPNet objective. In the Chignolin experiment, VAMPnet and DPNet objectives fail to converge, as shown in Table 2\. The only working example is DPNet-relaxed, and we demonstrated how LoRA can lead to a better fit of the neural networks to the trajectory data.
> * To further highlight LoRA's superior scalability and stability against DPNet-relaxed,  we conducted additional experiments on Chignolin dynamics (data split for seed $0$). We compared the test VAMP-E score on our LoRA variants against DPNet-relaxed by increasing the number of modes (**N**) and the SchNet node feature dimension (**H**), while also reducing the batch size (**B**).
>
>     The DPNet-relaxed baseline, previously considered a stable alternative, **completely failed to train under these conditions**, as indicated by the negative VAMP-E scores. In stark contrast, LoRA and its variants learned reliably and remained stable across all these challenging settings. This new experiment corroborates that LoRA solves the fundamental issues of numerical instability and resulting poor scalability in the prior work.
>
>     |(B,N,H)|SeqLoRA|JntLoRA|LoRA|DPNet-relaxed|DPNet-relaxed-Ctr|
>     |:-:|:-:|:-:|:-:|:-:|:-:|
>     |(384, 64, 128)|31.61|29.89|29.51|-9.76${}^2$|5.20${}^1$|
>     |(96, 64, 64)|25.83|23.82|25.51|4.75${}^1$|5.93${}^1$|
>     |(384, 64, 64)|26.66|27.25|24.11|-1.97${}^2$|4.72${}^1$|
>     |(384, 32, 128)|17.68|15.66|16.26|1.75|3.08|
>     |(96, 32, 64)|17.19|15.64|17.36|4.03${}^1$|6.44|
>     |(384, 32, 64)|16.69|17.02|15.79|4.36|4.91|
>     |(384, 16, 64)|9.73|8.79|8.80|3.05|4.19|
>     |(96, 16, 64)|10.22|8.80|8.52|2.74|4.40|
>
>     (VAMP-E scores are calculated on overall test data, not distinguishing folded/unfolded initialization.)
>     ${}^1$ During training, the VAMP-2 score exceeded the theoretical maximum (number of modes $N$). This is a strong indicator of numerical instability, suggesting a severe breakdown of orthogonality or a failure to learn diverse features.\\
>     ${}^2$ The negative VAMP-E score implies that the approximation error $\\|\\mathcal{K} \- \\hat{\\mathcal{K}}\\|$ is larger than the (Hilbert-Schmidt) norm of the true operator $\\|\\mathcal{K}\\|$ itself, rendering the approximation meaningless.

---

> > ### Author Response · Authors · 2025-08-06
> >
> > Thank you once again for your time and for your constructive review of our paper. We are writing to follow up on our rebuttal and to politely ask whether it has addressed the concerns you raised.
> >
> > In particular, we hope that our clarifications on the nesting techniques and the additional scalability experiments have helped provide a clearer picture of the advantages of LoRA and nesting. We aimed to respond directly to your comments on the consistency of nesting and the earlier lack of scalability evaluation.
> >
> > We would be very grateful for any further feedback you might have. We are eager to ensure that all your concerns are fully addressed before the discussion period ends.

---

### Note · Authors · 2025-08-13

We thank the reviewers and AC for discussions that sharpened our positioning and inspired new experiments.

Our work identifies a fundamental numerical instability in dominant Koopman learning objectives and resolves it with LoRA, a classical idea that has not previously been applied or validated in this setting. Our results show it is far from a naive application: in new Chignolin stress tests, LoRA remained stable where strong baselines collapsed, and the nesting technique further improved robustness.

We will integrate all feedback, and we believe the final version delivers a well-positioned and rigorously validated advance for Koopman learning, representing an important step toward more scalable data-driven Koopman learning methods.

---

### Decision · Program_Chairs · 2025-09-17

**Decision:**

Accept (poster)

**Comment:**

The paper addresses a key numerical instability in existing Koopman learning objectives and proposes LoRA to avoid ill-conditioned matrix inversions and for stable training. Reviewers raised concerns about the degree of novelty, clarity of nesting techniques, and reporting rigor, but after the extensive discussion phase were unanimously positive that this submission is a meaningful and practically important contribution. I agree with the eventually overall positive reception and recommend the authors follow through on their updates and promises from the discussion phase.